# FROM LARGE TO SMALL: TRANSFERRING CUDA OPTIMIZATION EXPERTISE VIA REASONING GRAPH

**Junfeng Gong**[1,2]   **Zhiyi Wei**[3]   **Junying Chen**[3]   **Cheng Liu**[1]*   **Huawei Li**[1]

[1]Institute of Computing Technology, Chinese Academy of Sciences
[2]University of Chinese Academy of Sciences
[3]School of Software Engineering, South China University of Technology
{gongjunfeng23s,liucheng}@ict.ac.cn

## ABSTRACT

Despite significant evolution of CUDA programming and domain-specific libraries, effectively utilizing GPUs with massively parallel engines remains difficult. Large language models (LLMs) show strong potential in generating optimized CUDA code from sequential code. However, using LLMs in practice faces two major challenges: cloud-based APIs pose risks of code leakage, and local deployment is often computationally expensive and inefficient. These drawbacks have spurred interest in small language models (SLMs), which are more lightweight and privacy-friendly. Encouragingly, recent studies show that SLMs can achieve performance comparable to LLMs on specific tasks. While SLMs can match LLMs on domain-specific tasks, their limited reasoning abilities lead to suboptimal performance in complex CUDA generation according to our experiments. To bridge this gap, we propose ReGraphT, a training-free, retrieval-augmented generation framework that transfers LLM-level reasoning to smaller models. ReGraphT organizes CUDA optimization trajectories into a structured reasoning graph, modeling the combined CUDA optimizations as state transitions, and leverages Monte Carlo Graph Search (MCGS) for efficient exploration. We also present a CUDA-specific benchmark with difficulty tiers defined by reasoning complexity to evaluate models more comprehensively. Experiments show that ReGraphT outperforms HPC-specific fine-tuned models and other retrieval-augmented approaches, achieving an average 2.33× speedup on CUDAEval and ParEval. When paired with DeepSeek-Coder-V2-Lite-Instruct and Qwen2.5-Coder-7B-Instruct, ReGraphT enables SLMs to approach LLM-level performance without the associated privacy risks or excessive computing overhead.

## 1 INTRODUCTION

The continuous performance improvement of NVIDIA GPUs (Dally et al., 2021; Lindholm et al., 2008; Nickolls & Dally, 2010; Owens et al., 2008) has solidified CUDA as a dominant programming model for high-performance computing tasks, including AI and scientific computing. However, writing efficient CUDA code that fully exploits the massively parallel processing capabilities of GPUs remains a significant challenge. To alleviate the burden of CUDA programming, prior research has proposed domain-specific libraries, programming frameworks, and even domain-specific languages (Brahmakshatriya & Amarasinghe, 2022; Tillet et al., 2019; Chen et al., 2018; Che et al., 2008; Bell & Hoberock, 2012; Hong et al., 2019). While these approaches significantly enhance productivity and deliver competitive performance, they often demand substantial engineering effort, are restricted to specific application domains, and suffer from compatibility issues with frequent NVIDIA software updates.

Recently, large language models (LLMs) have shown remarkable potential in code generation tasks(Qiu et al., 2020) across a wide range of programming languages—including Python, C/C++, Verilog, and even high-level synthesis code for FPGAs—demonstrating new opportunities for automatic CUDA code generation from sequential code (Li et al., 2023; Rozière et al., 2024; Luo

---

*Corresponding author.

et al., 2023; Zheng et al., 2023; Zhang et al., 2024; Xiong et al., 2025; Zhang et al., 2026). Encouraging progress has already been observed in this direction(Bendi-Ouis et al., 2025; Yan et al., 2024; Miranda et al., 2025). Nevertheless, deploying LLMs such as DeepSeek locally is highly resource-intensive due to their large-scale architecture. On the other hand, using cloud-based APIs raises concerns over potential code leakage and privacy violations. These limitations have fueled interest in small language models (SLMs), which are significantly more lightweight, support convenient local deployment, and mitigate privacy risks. Notably, recent studies have shown that SLMs can achieve performance on par with LLMs in certain domain-specific code generation tasks(Brown et al., 2020b).

Despite this promise, training SLMs from scratch remains extremely challenging due to limited training data and convergence difficulties. Consequently, fine-tuning has emerged as a practical means to produce compact, domain-specialized, and compute-efficient SLMs. For example, HPC-Coder-V1 and V2 leverage curated parallel-code datasets to fine-tune large LLMs and substantially improve their ability to generate high-performance parallel programs (Nichols et al., 2024b; Chaturvedi et al., 2025), while RLPF employs reinforcement learning to further align LLM outputs with performance objectives (Nichols et al., 2024c). However, we find that the efficacy of these fine-tuned SLMs degrades markedly on problems demanding deeper, multi-step reasoning. To quantify this effect, we sampled 20 benchmarks from ParEval and applied chain-of-thought(Wei et al., 2023) (CoT) prompting to both the 671B DeepSeek-R1 model and smaller 7B/14B code-specific SLMs. Figure 1 reports each model's average number of reasoning steps alongside the performance of the generated code. The results reveal that, while SLMs match LLMs on simpler tasks, they take significantly fewer reasoning steps and yield lower code quality on more complex benchmarks. This gap underscores the need for new techniques that can extend the reasoning capacity of lightweight models without sacrificing their deployment advantages.

In addition to fine-tuning, retrieval-augmented generation (RAG) is another widely adopted strategy for enhancing SLM performance by injecting external information directly into the model's context. Prior works such as EVOR and Repoformer (Su et al., 2024; Wu et al., 2024) have successfully applied RAG to general code generation tasks, demonstrating notable improvements in output quality, especially for code involving recurring patterns or known structures. However, while RAG effectively enriches contextual knowledge, it does not directly improve the model's reasoning capabilities. As a result, RAG-enhanced SLMs still struggle with generation tasks that require multi-step logical reasoning, leaving a critical gap in handling more complex coding problems.

To enhance the reasoning capabilities of SLMs in CUDA code generation, we propose ReGraphT, a training-free framework that augments SLMs with a structured reasoning process of CUDA-specific optimizations. ReGraphT leverages the reasoning strength of LLMs to collect step-by-step CUDA optimization trajectories, which are then aggregated into a unified CUDA reasoning graph. This graph captures the intermediate states and transitions involved in transforming sequential code into efficient CUDA implementations. ReGraphT formulates the CUDA code generation task for SLMs as a graph-based reasoning problem and incorporates Monte Carlo Graph Search (MCGS) to guide the search over the graph efficiently. In addition, to support systematic evaluation, we also introduce CUDAEval, a benchmark suite specifically designed to assess CUDA code generation. CUDAEval organizes tasks into multiple difficulty levels based on the complexity of their underlying reasoning trajectories, enabling fine-grained analysis of model performance across different levels of challenge.

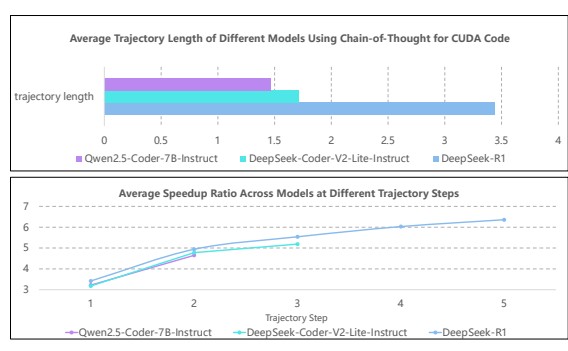

Figure 1: Average number of reasoning steps and the performance of the generated code with SLMs and LLMs.

Our contributions are summarized as follows:

- We propose ReGraphT, a novel, training-free framework designed to mitigate the limited reasoning ability of SLMs in CUDA code generation. ReGraphT employs a CUDA Reasoning Graph to encode optimization trajectories extracted from LLMs, thereby enabling SLMs to benefit from the rich multi-step reasoning encoded by larger models. The framework is open sourced on GitHub[1].

- We formulate CUDA code generation as a graph-based state transition problem and apply Monte Carlo Graph Search (MCGS) to efficiently navigate the CUDA Reasoning Graph. This formulation enables effective decision-making at each optimization stage, enhancing the quality of generated CUDA code.

- We design CUDAEval, a CUDA-specific benchmark suite that categorizes code generation tasks into levels of reasoning difficulty. Experimental results show that ReGraphT significantly improves the reasoning and code generation performance of SLMs, narrowing the gap between lightweight and large models in CUDA optimization tasks.

## 2 RELATED WORK

To support efficient CUDA programming, NVIDIA has developed a suite of CUDA Toolkit Libraries such as cuBLAS, cuDNN, and cuFFT (NVIDIA Corporation, 2023a;b;c), which offer optimized implementations for common parallel kernels. For effective CUDA code generation for domain-specific applications such as AI and multimedia, several compilation-based methods (NVIDIA Corporation, 2023d; Chen et al., 2018; Ragan-Kelley et al., 2013; Tillet et al., 2019) have been proposed.

Beyond compiler-based methods, recent research has explored improving code generation via language models. LLMs have demonstrated strong capabilities in code generation across various domains, including general-purpose programming, hardware design, and high-performance computing. However, practical deployment of LLMs presents two major challenges: cloud-based APIs raise concerns over potential code leakage, while local deployment is often computationally expensive and inefficient. These limitations have driven interest in small language models (SLMs), which offer lightweight alternatives suitable for local use.

Supervised fine-tuning (SFT) has shown promise in domain-specific tasks—e.g.(Fatemi & Hu, 2023) fine-tunes smaller LLMs for financial sentiment analysis with competitive results. HPC-Coder (Nichols et al., 2024b; Chaturvedi et al., 2025) enhances LLM performance in generating high-performance computing (HPC) code through fine-tuning with high-quality synthesized datasets. However, SFT has limited effectiveness in boosting multi-step reasoning capabilities in SLMs and often suffers from poor generalization (Ghosh et al., 2024). Knowledge distillation from LLM-generated synthetic data has emerged as an alternative for improving SLM reasoning ability (DeepSeek-AI et al., 2025; Wang et al., 2025). While effective, this approach relies heavily on carefully crafted data recipes, making the distillation process challenging and sensitive to dataset composition. The generalization of LLMs can also be improved by injecting relevant external knowledge through RAG, but it may also introduce hallucinations or irrelevant information (Sun et al., 2025; Gao et al., 2024). It becomes problematic particularly for CUDA code generation which typically combines multiple optimization techniques.

## 3 THE PROPOSED REGRAPHT FRAMEWORK

To address the reasoning limitations of SLMs in CUDA code generation, we propose ReGraphT, a lightweight, training-free framework that augments SLMs with structured reasoning guidance. As shown in Figure 2, ReGraphT first leverages LLMs to extract multi-step CUDA optimization trajectories from sequential code, organizing them into a CUDA Reasoning Graph. This graph encodes the step-by-step transformation paths and serves as a reasoning scaffold for SLMs. Then, it frames CUDA optimization as a graph traversal problem and applies Monte Carlo Graph Search (MCGS) for guided exploration, which enables SLMs to generate higher-quality CUDA code with improved multi-step reasoning capabilities.

---

[1]`https://github.com/blacknickwield/ReGraphT`

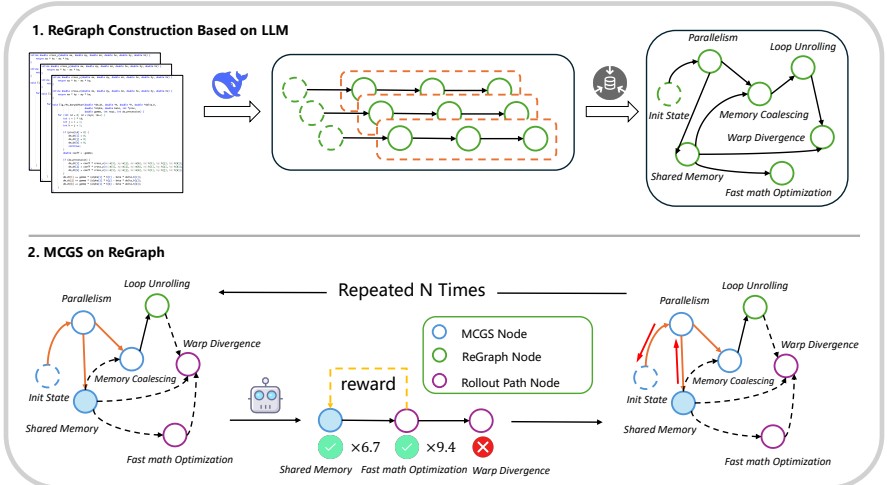

Figure 2: **Overview of the proposed ReGraphT framework.**

## 3.1 REASONING GRAPH (REGRAPH) CONSTRUCTION

CoT (Wei et al., 2023) improves the ability of LLMs to perform complex reasoning through a series of intermediate reasoning steps, allowing SLMs with limited intelligence to emulate the reasoning process of LLMs, boosting their performance on tasks involving planning and reasoning. Figure 3 shows how CoT works in CUDA optimization and produces a reasoning trajectory. To efficiently utilize the intermediate reasoning traces generated by LLMs, we propose to organize the CUDA optimization expertise in a novel graph structure called ReGraph. Prior to discussing the construction of ReGraph, we formally present its definition.

**Definition 1.** *ReGraph can be defined as $\mathcal{G} = (\mathcal{V}, \mathcal{E})$, referring to a directed graph-based abstraction derived from CUDA optimization expertise. In ReGraph $\mathcal{G}$, each $v \in \mathcal{V}$ represents an identified CUDA optimization technique, each $u \in \mathcal{E}$ represents the link between two optimization methods.*

According to Definition 1, ReGraph adopts a directed graph representation that permits cycles. In ReGraphT, we formulate CUDA code optimization as a state transition process on the graph. As all optimization processes start with sequential codes, there exists an initial state $v_{init}$ in $\mathcal{G}$, which stands for the starting point of optimization. At initialization, ReGraph $\mathcal{G}$ consists exclusively of vertex $v_{init}$, devoid of any edges. Building upon this, ReGraph completes the construction of the entire graph by merging CUDA optimization trajectories. Algorithm 1 illustrates the complete process of ReGraph construction.

To acquire CUDA optimization trajectories, we prompt LLM to perform CUDA optimization step by step, thus yielding a CUDA optimization trajectory. For each intermediate step of the trajectory, we instruct LLM to provide CUDA optimization method used, optimized CUDA code and corresponding reasoning process. Due to the stochastic nature of LLM outputs, identical CUDA optimization methods may be expressed differently by the LLM during different steps. Therefore, during the construction process, ReGraphT systematically records the existing CUDA optimization methods and prompts LLM to consolidate the current optimization trajectory with documented methods, thereby ensuring the consistency in the representation of each optimization method.

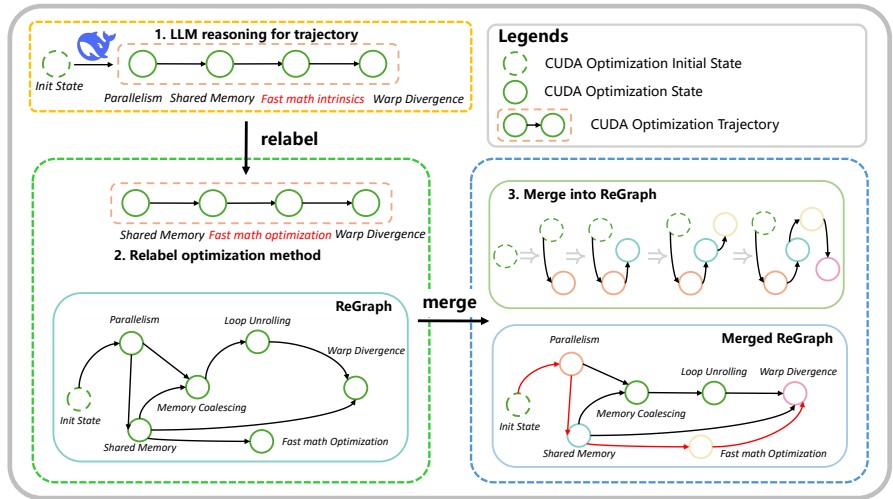

Figure 3: **ReGraph construction based on LLM optimization trajectory.**

---

**Algorithm 1:** ReGraph Generation

---

**Require:** Sequential code dataset, D
**Require:** Large language model, LLM
**Output:** ReGraph $\mathcal{G} = (\mathcal{V}, \mathcal{E})$

1 Initialize the set of CUDA optimization methods $\mathcal{O} = \{\}$
2 Initialize the nodes of CUDA Reasoning Graph $\mathcal{V} = \{v_{init}\}$
3 Initialize the edges of CUDA Reasoning Graph $\mathcal{E} = \{\}$

4 **for** $k \in D$ **do**
5 $\quad$ — *Trajectory of CUDA optimization* —
6 $\quad$ $\tau \leftarrow LLM(k)$
7 $\quad$ $\tau^{'} \leftarrow relabel(LLM, \tau, \mathcal{O})$
8 $\quad$ — *CUDA Reasoning Graph merge* —
9 $\quad$ $s \leftarrow v_{init}$
10 $\quad$ **for** $e \in \tau^{'}$ **do**
11 $\quad\quad$ **Get** optimization method $o \leftarrow Method(e)$
12 $\quad\quad$ **if** $o \in \mathcal{O}$ **then**
13 $\quad\quad\quad$ **Find** the node $v$ corresponding to $o$
14 $\quad\quad\quad$ **if** $v \in Succ(s)$ **then**
15 $\quad\quad\quad\quad$ **Find** the edge $u$ between $s$ and $v$
16 $\quad\quad\quad\quad$ **Append** optimization example $e$ to $u$
17 $\quad\quad\quad$ **else**
18 $\quad\quad\quad\quad$ $u \leftarrow Edge(s, v)$
19 $\quad\quad\quad\quad$ **Append** optimization example $e$ to $u$
20 $\quad\quad\quad\quad$ **Append** new edge $u$ to $\mathcal{E}$
21 $\quad\quad\quad$ $s \leftarrow v$
22 $\quad\quad$ **else**
23 $\quad\quad\quad$ $v \leftarrow Node(o)$
24 $\quad\quad\quad$ $u \leftarrow Edge(s, v)$
25 $\quad\quad\quad$ **Append** optimization example $e$ to $u$
26 $\quad\quad\quad$ **Append** new node $v$ to $\mathcal{V}$
27 $\quad\quad\quad$ **Append** new edge $u$ to $\mathcal{E}$
28 $\quad\quad\quad$ $s \leftarrow v$

29 **return** *CUDA Reasoning Graph* $\mathcal{G} = (\mathcal{V}, \mathcal{E})$

---

After the CUDA optimization trajectory was produced, ReGraphT merges the new trajectory into ReGraph. As mentioned in Definition 1, CUDA Reasoning Graph composes of CUDA optimization method nodes and edges representing the link between different optimization methods. From this perspective, a CUDA optimization trajectory can be regarded as a specific state transition trajectory. The detailed state transition process is described by Algorithm 1 on lines 8 - 29. For the current trajectory $\tau'$, state $s$ is initialized to $v_{init}$. For each intermediate step in $\tau'$, ReGraphT determines whether its corresponding optimization method is already incorporated in CUDA Reasoning Graph at first. If incorporated, it processes separately based on whether the state transition it represents exists (lines 13 - 22); otherwise, adds the method to the CUDA Reasoning Graph (lines 24 - 29). Afterwards, the current state $s$ will be updated, which means moving to the node corresponding to the optimization method. More detailed construction steps of ReGraph are provided in Appendix.

## 3.2 REASONING GRAPH (REGRAPH) EXPLORATION

Once ReGraph is constructed, ReGraphT leverages it to achieve the transfer of reasoning capabilities to SLMs via graph search, which means treating CUDA optimization as state transitions on ReGraph and determining the next optimization method used following a predefined strategy. A feasible search strategy is to enumerate all possible combinations of CUDA optimization methods based on ReGraph. Specifically, for each optimization state, all subsequent viable optimization methods are attempted until no more methods can be applied. However, despite the pruning of paths due to certain optimization methods being inapplicable, the time complexity of the enumeration search can still reach $O(n^k)$, where $n$ is the number of nodes in CUDA Reasoning Graph and $k$ represents the average of subsequent optimization methods for a node.

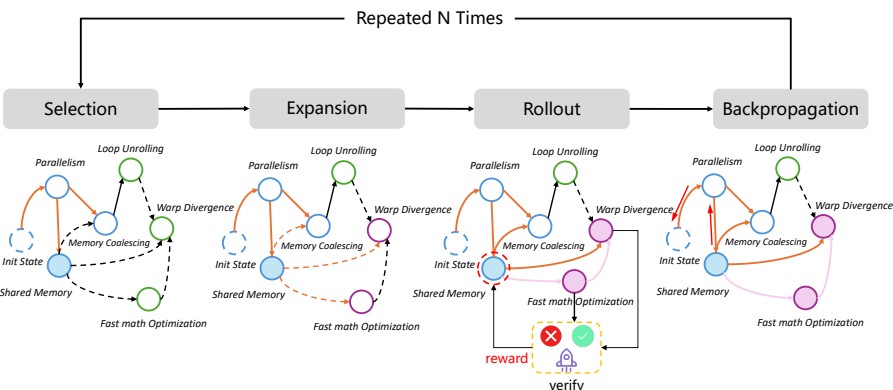

Figure 4: **An overview of MCGS on ReGraph.**

To tackle the complexity of enumeration search, we propose Monte Carlo Graph Search (MCGS), combining Monte Carlo Tree Search (MCTS) with ReGraph, leveraging rollout feedback from future states to inform the subsequent choice of optimization methods. To enable MCTS on the graph structure, we introduce some adaptations to the standard MCTS. As shown in Figure 4, we customize the key operations of MCGS on CUDA Reasoning Graph as follows:

**Selection:** As MCGS progresses, nodes and edges from ReGraph are incrementally added to form a new graph, which stands as a sub-graph of ReGraph. In the current iteration, MCGS select nodes from the existing graph based on UCB (Upper Confidence Bound):

$$\text{P-UCB}(s) = Q(s) + \sqrt{\frac{2\ln(N(s'))}{N(s)}} \tag{1}$$

**Expansion:** Unlike MCTS which decomposes problems at thought-level(Chen et al., 2024; Xie et al., 2024; Li et al., 2024; Hu et al., 2025), since MCGS method operates on a fixed graph ReGraph, the action space in each expansion step is also fixed—specifically, the successor nodes of the current optimization state within the ReGraph. If the node selected in the previous step has not been visited before, all successors will be expanded in MCGS to extend the entire search scope.

**Rollout:** A rollout refers to simulating from the current state to evaluate it. Unlike MCTS which performs estimations on the tree, ReGraph contains cycles, which may cause simulations to fail to terminate. To facility the problem, we made certain adjustments to the simulation strategy.

- To avoid repeated visits to the same node, we incorporated a regularization term based on the current visit count in the simulation, balancing exploitation and exploration more effectively than standard $\epsilon$-greedy:

$$\pi(a|s) = \begin{cases} \arg\max_a \left[ Q(s,a) - \lambda N(s,a) \right] & \text{with probability } 1 - \epsilon \\ \text{random action} & \text{with probability } \epsilon \end{cases} \tag{2}$$

- We set a maximum step limit for each rollout to prevent non-termination. What's more, it will also terminate if the optimization fails at any node.

CUDA optimization requires error-free compilation while maximizing performance. As a result, at each step of the rollout process, the optimized CUDA code undergoes compilation verification, functional validation, and performance benchmarking, yielding the following hierarchical reward design:

$$\text{reward} = \begin{cases} -1, & \text{if } 0 \leq v^{\text{test}} < 1, \\ \text{speedup} - 1, & \text{if } v^{\text{test}} = 1 \text{ and speedup} < 1, \\ \text{speedup}, & \text{if } v^{\text{test}} = 1 \text{ and speedup} \geq 1. \end{cases} \tag{3}$$

In MCGS, each node can be treated as a terminal state, generating the final optimized code. The rollout's final reward is defined as the maximum reward observed on its trajectory.

**Backpropagation** To enable the rewards obtained in the current iteration to guide subsequent processes, MCGS backpropagates the rewards along all nodes traversed in the selection path, updating their Q-values. After backpropagation, MCGS progresses to the next iteration.

## 4 THE PROPOSED CUDA EVALUATION BENCHMARK (CUDAEVAL)

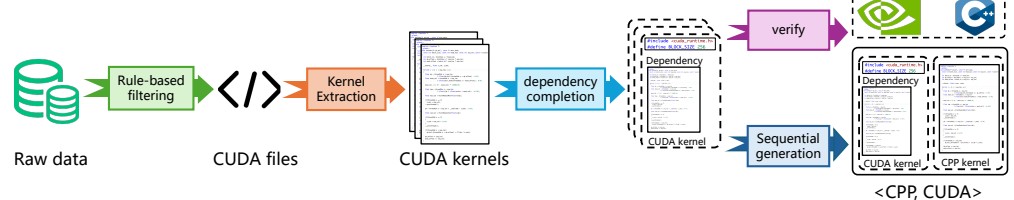

Figure 5: **CUDAEval curation process.**

Existing benchmarks such as HumanEval and MBPP (Chen et al., 2021; Austin et al., 2021) primarily evaluate the functional correctness of LLM-generated code. ParEval (Nichols et al., 2024a), while designed for assessing parallel code generation, focuses mainly on various parallel paradigms and includes only 60 CUDA-specific instances—limited in both scale and diversity. Moreover, it lacks a fine-grained classification scheme that reflects the complexity of real-world CUDA development. To address these limitations, we present CUDAEval, a dedicated benchmark for evaluating LLM performance in CUDA code generation across varying levels of reasoning complexity. Unlike prior benchmarks that start from sequential code, CUDAEval is built from real-world CUDA files. Specifically, we sample 10K CUDA files from the Stack_v2_cuda_hip dataset, which comprises 21.7K CUDA files collected from practical development scenarios.

The benchmark is curated by first applying heuristic rules (e.g., filtering files with local headers) to remove incomplete or unbuildable samples, followed by LLM-based extraction of CUDA kernels and completion of missing dependencies. LLMs also generate corresponding CPU serial code and

driver functions, with correctness verified via compilation and execution of both parallel and serial versions. Only `<C++, CUDA>` pairs that pass build and output consistency checks are retained.

After validation, we obtain 3,126 high-quality CUDA code pairs. Using DeepSeek-R1 (DeepSeek-AI et al., 2025), we derive optimization trajectories and classify each sample into one of three difficulty levels based on reasoning complexity. Specifically, samples with trajectory lengths of 1–2 are assigned to the easy-tier, those with lengths of 3–5 to the medium-tier, and those with longer trajectories to the hard-tier. Under this definition, the dataset comprises 1,783 easy, 791 medium, and 552 hard instances. We selected 10% of the tasks for final evaluation while the left are used for ReGraph construction. To further increase the challenge of the benchmark, we deliberately included a relatively larger proportion of the harder samples. As a result, the final CUDAEval contains 313 evaluation tasks, distributed across 106 easy, 105 medium, and 102 hard samples. Full details of CUDAEval pipeline are provided in the Appendix.

# 5 EXPERIMENTS

We conduct our experiments on a single A100-80GB with Intel(R) Xeon(R) Platinum 8358P CPU @ 2.60GHz. For inference, we deploy LLMs using vLLM(Kwon et al., 2023) under BF16 precision.

## 5.1 EXPERIMENT SETUPS
**Benchmarks:** We have both CUDAEval and an established benchmark ParEval(Nichols et al., 2024a) to evaluate the CUDA generation performance. ParEval covers 12 different computational problems and 7 parallel models, but only 60 problems are available for CUDA code generation.

**Baselines:** We compare ReGraphT with prior prompting and RAG methods. For prompting methods, we compare it with standard(Brown et al., 2020a) and CoT Prompting(Wei et al., 2023). For RAG methods, since there are no RAG methods specifically designed for CUDA optimization, we construct several RAG variants with the same CUDA optimization corpus used in ReGraphT. Specifically, we adopted a code similarity-based retrieval approach as the RAG baseline, which employ CodeBERTScore(Zhou et al., 2023) as the embedding model to retrieve relevant CUDA optimization examples based on embedding similarity. In addition to this similarity-based RAG baseline, we further include two stronger baselines—RethinkMCTS(Li et al., 2024) and MCTS-RAG(Hu et al., 2025)—to provide a more comprehensive comparison within the RAG family.

**Hyperparameters:** ReGraphT and ReGrapht-MCGS are evaluated under the same search budgets of 200. ReGraphT adopts a random sampling with max attempts of 5. The varying rollout configuration $N$ in Figure 4 is set to 10. More details about Hyperparameter settings are in Appendix.

**Metrics:** To quantify the correctness of generated CUDA code, we adopt pass@k introduced in (Chen et al., 2021), while for optimization performance, we use speedup@k(Nichols et al., 2024a) to evaluate the performance improvement over the original sequential code.

## 5.2 EXPERIMENT RESULTS

Table 1 presents the CUDA optimization performance under different methods with various code-specific SLMs DeepSeek-Coder-V2-Lite-Instruct, Qwen2.5-Coder-7B-Instruct(Yang et al., 2024; Hui et al., 2024), and HPC-Coder-V2(Chaturvedi et al., 2025), and the SOTA general LLMs DeepSeek-V3-0324(DeepSeek-AI, 2024) and DeepSeek-R1(DeepSeek-AI et al., 2025). In CUDAEval, ReGraphT-MCGS achieves 73.3% in pass@k on average with three code-specific SLMs, surpassing by +11.0%, +9.0% and +6.8% compared to Standard, CoT and RAG in pass@1, +9.2%, +7.1% and +5.5% in pass@10. While ensuring the correctness of generated code, ReGraphT-MCGS also demonstrates superior quality in CUDA code generation, achieving at least ×1.84 speedup in speedup@1 and ×1.83 in speed@10 compared to other baselines. For the search-based variants RethinkMCTS and MCTS-RAG, although they achieve some improvement over other baselines, they still fall short of our ReGraphT-MCGS method. On the overall more challenging ParEval, ReGraphT-MCGS also demonstrates outstanding performance.

Beyond baseline comparisons, we further verify the efficacy of the MCGS strategy on ReGraph. According to Table 1, under the fixed search budgets of 200, ReGraph-MCGS achieves higher performance to ReGraphT, with +2.2% pass@n, +1.33% speedup@n increase on average in CUDAEval and +1.7% pass@n, +0.34% speedup@n in ParEval.Our experiments demonstrate that, under

Table 1: CUDA generation performance on CUDAEval and ParEval benchmarks.

| Model | Method | CUDAEval | | | | ParEval | | | |
|---|---|---|---|---|---|---|---|---|---|
| | | pass@n | | speedup@n | | pass@n | | speedup@n | |
| | | pass@1 | pass@10 | speedup@1 | speedup@10 | pass@1 | pass@10 | speedup@1 | speedup@10 |
| DeepSeek-Coder-V2-Lite-Instruct | Standard | 61.7 | 63.9 | 6.54 ± 0.74 | 6.76 ± 0.71 | 40.0 | 42.1 | 4.61 ± 0.69 | 4.82 ± 0.67 |
| | CoT(Wei et al., 2023) | 64.9 | 67.4 | 7.23 ± 0.72 | 7.39 ± 0.70 | 43.3 | 43.9 | 4.94 ± 0.67 | 4.97 ± 0.67 |
| | RAG(Zhou et al., 2023) | 68.1 | 70.9 | 7.86 ± 0.79 | 7.89 ± 0.76 | 48.3 | 48.7 | 5.35 ± 0.74 | 5.34 ± 0.73 |
| | RethinkMCTS(Li et al., 2024) | 68.7 | | 7.76 ± 0.93 | | 50.0 | | 5.12 ± 0.89 | |
| | MCTS-RAG(Hu et al., 2025) | 71.6 | | 8.09 ± 0.82 | | 51.7 | | 5.78 ± 0.84 | |
| | ReGraphT | 73.2 | | 13.02 ± 0.85 | | 51.7 | | 10.06 ± 0.79 | |
| | ReGraphT-MCGS | **75.1** | | **14.46 ± 0.87** | | **55.0** | | **10.78 ± 0.82** | |
| Qwen2.5-Coder-7B-Instruct | Standard | 61.0 | 63.6 | 6.34 ± 0.77 | 6.32 ± 0.74 | 38.3 | 39.2 | 4.33 ± 0.70 | 4.51 ± 0.68 |
| | CoT(Wei et al., 2023) | 62.3 | 64.2 | 6.31 ± 0.76 | 6.31 ± 0.75 | 35.0 | 38.8 | 4.30 ± 0.75 | 4.47 ± 0.72 |
| | RAG(Zhou et al., 2023) | 66.5 | 67.1 | 7.09 ± 0.86 | 7.24 ± 0.82 | 45.0 | 45.3 | 5.17 ± 0.79 | 5.20 ± 0.73 |
| | RethinkMCTS(Li et al., 2024) | 66.5 | | 6.84 ± 1.02 | | 48.3 | | 5.09 ± 0.86 | |
| | MCTS-RAG(Hu et al., 2025) | 68.4 | | 7.51 ± 0.94 | | 50.0 | | 5.56 ± 0.84 | |
| | ReGraphT | 69.6 | | 12.89 ± 0.85 | | **51.7** | | **10.11 ± 0.75** | |
| | ReGraphT-MCGS | **72.2** | | **14.31 ± 0.81** | | 50.0 | | 10.02 ± 0.75 | |
| HPC-Coder-V2 | Standard | 64.2 | 64.9 | 6.48 ± 0.68 | 6.53 ± 0.65 | 36.7 | 37.1 | 4.47 ± 0.72 | 4.59 ± 0.70 |
| | CoT(Wei et al., 2023) | 65.8 | 67.1 | 6.93 ± 0.73 | 7.02 ± 0.70 | 30.0 | 40.7 | 4.73 ± 0.72 | 4.86 ± 0.68 |
| | RAG(Zhou et al., 2023) | 64.8 | 65.5 | 6.44 ± 0.73 | 6.50 ± 0.69 | 38.3 | 39.9 | 4.51 ± 0.69 | 4.57 ± 0.69 |
| | RethinkMCTS(Li et al., 2024) | 67.1 | | 7.48 ± 1.26 | | 41.3 | | 5.01 ± 0.93 | |
| | MCTS-RAG(Hu et al., 2025) | 68.4 | | 7.15 ± 0.97 | | 40.7 | | 4.92 ± 0.88 | |
| | ReGraphT | 70.6 | | 13.26 ± 0.84 | | 50.0 | | 10.21 ± 0.80 | |
| | ReGraphT-MCGS | **72.5** | | **14.39 ± 0.83** | | **53.3** | | **10.61 ± 0.82** | |
| DeepSeek-R1-Distill-Qwen-7B | Standard | 63.9 | 64.9 | 7.52 ± 0.67 | 7.59 ± 0.71 | 43.9 | 45.3 | 5.08 ± 0.65 | 5.17 ± 0.61 |
| | CoT(Wei et al., 2023) | 67.1 | 69.3 | 8.16 ± 0.69 | 8.16 ± 0.69 | 48.1 | 50.0 | 5.40 ± 0.65 | 5.54 ± 0.59 |
| | RAG(Zhou et al., 2023) | 66.5 | 68.4 | 8.43 ± 0.66 | 8.65 ± 0.62 | 48.3 | 50.0 | 5.73 ± 0.67 | 5.81 ± 0.67 |
| | RethinkMCTS(Li et al., 2024) | 71.6 | | 7.92 ± 0.89 | | 51.7 | | 6.34 ± 0.82 | |
| | MCTS-RAG(Hu et al., 2025) | 71.6 | | 8.06 ± 0.91 | | 53.3 | | 6.57 ± 0.85 | |
| | ReGraphT | 75.8 | | 14.15 ± 0.77 | | 55.0 | | 10.92 ± 0.72 | |
| | ReGraphT-MCGS | **76.4** | | **14.72 ± 0.73** | | 55.0 | | **11.25 ± 0.67** | |
| DeepSeek-V3-0324 | Standard | 79.6 | 80.8 | 18.71 ± 0.64 | 18.86 ± 0.59 | 63.3 | 63.8 | 11.40 ± 0.60 | 10.99 ± 0.61 |
| | CoT(Wei et al., 2023) | 80.2 | 81.5 | 18.58 ± 0.57 | 18.45 ± 0.55 | 61.7 | 62.1 | 11.83 ± 0.62 | 11.77 ± 0.59 |
| DeepSeek-R1 | Standard | 80.5 | 81.5 | 19.02 ± 0.73 | 19.45 ± 0.71 | 58.3 | 58.2 | 11.52 ± 0.69 | 11.57 ± 0.75 |
| | CoT(Wei et al., 2023) | 82.1 | 83.1 | 19.14 ± 0.75 | 19.62 ± 0.70 | 63.3 | 63.6 | 12.09 ± 0.68 | 12.13 ± 0.68 |

the same search budget constraints, ReGraphT-MCGS enables a more efficient exploration over ReGraph compared to ReGraphT.

Table 2: CUDA Generation Performance Across Three Difficulty Levels in CUDAEval.

| Model | Method | pass@n | | | speedup@n | | |
|---|---|---|---|---|---|---|---|
| | | easy | medium | hard | easy | medium | hard |
| DeepSeek-Coder-V2-Lite-Instruct | Standard | 81.1 | 65.7 | 44.1 | 8.90 ± 0.69 | 6.32 ± 0.72 | 3.34 ± 0.72 |
| | CoT | 86.8 | 68.6 | 46.1 | 9.65 ± 0.67 | 6.51 ± 0.71 | 4.31 ± 0.72 |
| | RAG | **91.5** | 73.3 | 47.1 | 10.13 ± 0.78 | 6.98 ± 0.74 | 4.82 ± 0.73 |
| | ReGraphT | 90.6 | 76.2 | 52.0 | 15.86 ± 0.80 | 12.38 ± 0.87 | 8.84 ± 0.86 |
| | ReGraphT-MCGS | 90.6 | **79.0** | **54.9** | **17.82 ± 0.80** | **13.79 ± 0.80** | **9.69 ± 0.82** |
| Qwen2.5-Coder-7B-Instruct | Standard | 81.1 | 65.7 | 43.1 | 8.51 ± 0.73 | 5.62 ± 0.76 | 3.14 ± 0.80 |
| | CoT | 82.1 | 66.7 | 43.1 | 8.47 ± 0.75 | 5.54 ± 0.79 | 3.46 ± 0.78 |
| | RAG | 85.8 | 69.5 | 45.1 | 9.54 ± 0.80 | 6.23 ± 0.81 | 4.29 ± 0.84 |
| | ReGraphT | 85.8 | 73.3 | 49.0 | 15.67 ± 0.81 | 12.26 ± 0.86 | 8.80 ± 0.88 |
| | ReGraphT-MCGS | **88.7** | **76.2** | **51.0** | **17.48 ± 0.83** | **13.64 ± 0.83** | **9.61 ± 0.84** |
| HPC-Coder-V2 | Standard | 83.0 | 66.7 | 44.1 | 8.78 ± 0.63 | 6.17 ± 0.67 | 2.69 ± 0.66 |
| | CoT | 86.8 | 68.6 | 45.1 | 9.31 ± 0.72 | 6.13 ± 0.69 | 3.83 ± 0.70 |
| | RAG | 81.1 | 66.7 | 44.1 | 8.82 ± 0.69 | 6.22 ± 0.69 | 3.08 ± 0.68 |
| | ReGraphT | 88.7 | 71.4 | 51.0 | 16.43 ± 0.82 | 12.45 ± 0.85 | 8.70 ± 0.83 |
| | ReGraphT-MCGS | **90.6** | **74.3** | **52.0** | **17.66 ± 0.81** | **13.58 ± 0.86** | **9.66 ± 0.83** |
| DeepSeek-V3-0324 | Standard | 93.4 | 87.6 | 60.8 | 23.23 ± 0.59 | 18.54 ± 0.58 | 12.36 ± 0.61 |
| | CoT | 93.4 | 88.6 | 61.8 | 22.66 ± 0.52 | 18.38 ± 0.56 | 11.94 ± 0.57 |
| DeepSeek-R1 | Standard | 94.3 | 87.6 | 61.8 | 24.01 ± 0.68 | 18.73 ± 0.71 | 13.26 ± 0.71 |
| | CoT | 95.3 | 89.5 | 63.7 | 24.24 ± 0.69 | 18.96 ± 0.72 | 13.40 ± 0.70 |

To analyze the impact of reasoning capability on CUDA code optimization performance, we further investigated the relationship between the length of reasoning trajectories and corresponding performance based on performances across different difficulty levels in CUDAEval. As show in Table 2, ReGraph demonstrates varying performance across different difficulty levels. On the easy level which requires minimal reasoning, ReGraph shows no significant gap compared to other baselines, and occasionally underperforms CoT and RAG approaches. However, as task difficulty escalates to medium and hard levels demanding more advanced reasoning, ReGraph begins to demonstrate marked advantages over alternative methods.

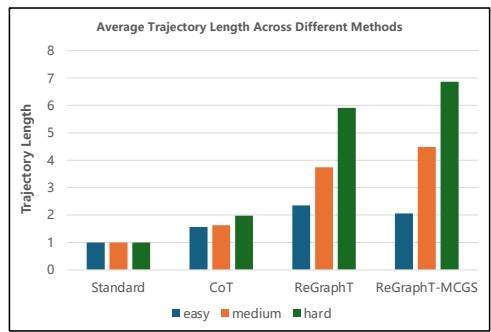 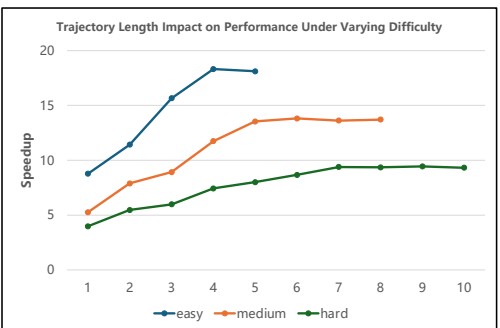

Figure 6: **Normalized performance of the generated CUDA code under various difficulty levels**

To further demonstrate ReGraphT's enhancement of SLMs' reasoning capabilities in CUDA generation tasks, we perform a deeper examination regarding the correlation between the complexity of reasoning trajectories and optimization performance. As observed in Figure 6, limited by the reasoning capacity of SLMs, the length of CoT reasoning trajectories exhibits minimal variation across difficulty levels, while for ReGraphT series, the difference in the average length of reasoning trajectories between the easy and hard difficulty levels can reach up to 4.8, thus demonstrating that ReGraph can boost SLMs reasoning in CUDA generation process. What's more, in comparison to ReGraphT, ReGraphT-MCGS exhibits longer average reasoning trajectories, highlighting its advantage in search efficiency. In addition to reasoning boosting, Figure 6 further demonstrates the role of reasoning in enhancing performance for CUDA optimization tasks. From the figure, we observe a positive correlation between CUDA generation performance and reasoning chain length across all difficulty levels, until the reasoning steps reach a certain threshold. Notably, different difficulty tasks exhibit distinct thresholds, which generally show positive correlation with task difficulty. Beyond the threshold, the performance gap becomes statistically insignificant.

## 6 CONCLUSION

In this paper, we propose ReGraphT, a training-free framework which transfers the CUDA optimization reasoning capability of LLMs to SLMs via Reasoning Graph. According to experiment results, ReGraphT has demonstrated significant effectiveness in enhancing SLM's reasoning capability for CUDA-generated content and improving generation quality. This work demonstrates that the proposed reasoning graph can transfer the reasoning capability of LLMs to SLMs effectively and ReGraphT can be potentially applied to more code generation scenarios that require complex or long reasoning procedures. We will investigate this approach in our future work.

## ACKNOWLEDGMENTS

This work was partially supported by the Strategic Priority Research Program of the Chinese Academy of Sciences (Grant No. XDB0660103), the State Key Laboratory of Processors, Institute of Computing Technology, Chinese Academy of Sciences (Grant No. CLQ202403), the Guangdong Basic and Applied Basic Research Foundation (Grant No. 2024A1515030128), and the National Natural Science Foundation of China (Grant No. U2570205).

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

## A    CUDAEVAL CURATION

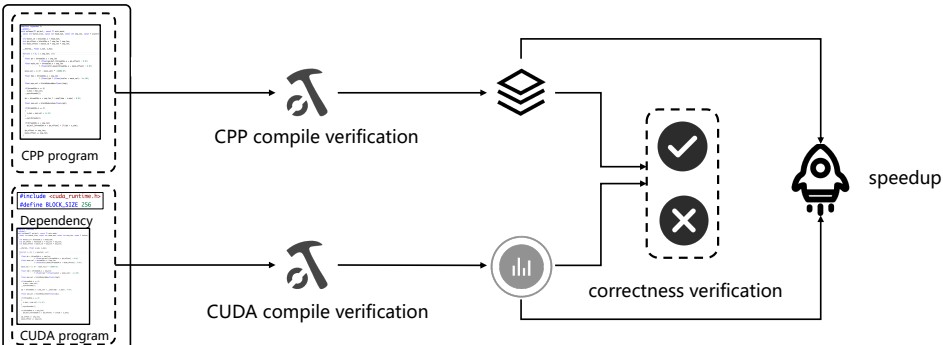

Figure 7: **CUDAEval verification process.**

**Rule-based Preprocession** The main goal of our filtering rules is to reduce the cost of LLM api calling. To achieve this goal, we designed the following heuristic filtering rules:

- Remove CUDA files that include local header files

- Retain only files where code functions contain between 50 to 500 lines.

- Filter out files containing dependencies on CUDA third-party libraries (including but not limited to cuDNN, cuBLAS).

**Kernel Extraction and Dependency Completion** After heuristic rule filtering, we employ prompts to instruct the LLM to extract CUDA kernels and their corresponding dependencies from the remaining files. Since these CUDA files were collected from repositories, the extracted kernels may still exhibit issues such as missing macro definitions, absent class definitions, and similar deficiencies.

To address the aforementioned issues, we employ the LLM to attempt dependency completion for these kernels. The specific prompt used for this purpose is illustrated in the accompanying figure.

**Sequential Code and Driver Generation** After kernel extraction and dependency completion, we generate their corresponding serial codes based on the parallel codes and construct the main functions to call them respectively in preparation for the subsequent verification phase.

**Verification Pipeline** To maintain data correctness and improve quality, we implemented comprehensive validation, specifically examining both accuracy and performance metrics. First, we will compile and verify the two code segments separately. After confirming successful compilation, we execute both programs using the same test data and compare their outputs to validate correctness. Once correctness is ensured, we evaluate their runtime performance and select the code that demonstrates acceleration effects. The complete verification process is illustrated in Figure 7.

## B  REGRAPH CONSTRUCTION

**LLM reasoning for CUDA Optimization trajectory** As shown in Figure 8., We instruct LLM to carry out CUDA optimizations stepwise in order to derive optimization trajectories using prompt G.3. However, LLM lacks the ability to verify either the correctness or the effectiveness of its own CUDA optimizations. As a result, prior to merging the CUDA optimization trajectories into ReGraph, we need to validate the results and performance of every optimization step. The verification approach follows the same methodology in Figure 7.

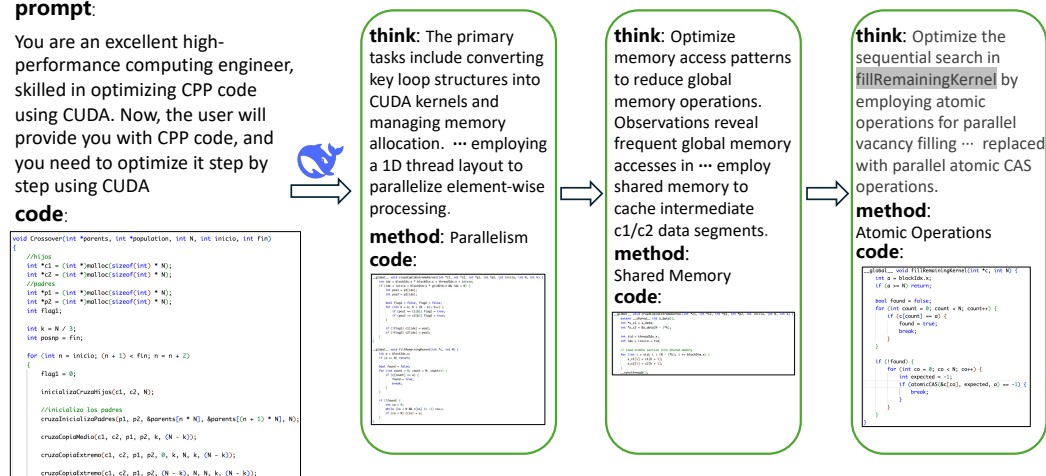

Figure 8: **Process of LLM reasoning for CUDA optimization.**

**Optimization trajectory relabel** Following verification, it remains necessary to employ the LLM to re-annotate every step in the optimization trajectory according to established CUDA optimization techniques. Specifically, using the prompt illustrated in G.4, we instruct the LLM to determine whether each optimization method employed in the current trajectory aligns with existing CUDA optimization methods. When a match is found, the corresponding optimization method is renamed accordingly.

## C  ANALYSIS OF REGRAPH DISTRIBUTION

Figure 9 illustrates the relationship between the distribution of ReGraph and the number of samples used for its construction. Our experiments show that the reasoning graph converges after ap-

proximately 500 samples. This is reasonable given that the space of effective CUDA optimization strategies is inherently limited. Therefore, ReGraph is able to capture most of the commonly used optimizations using only a limited number of samples.

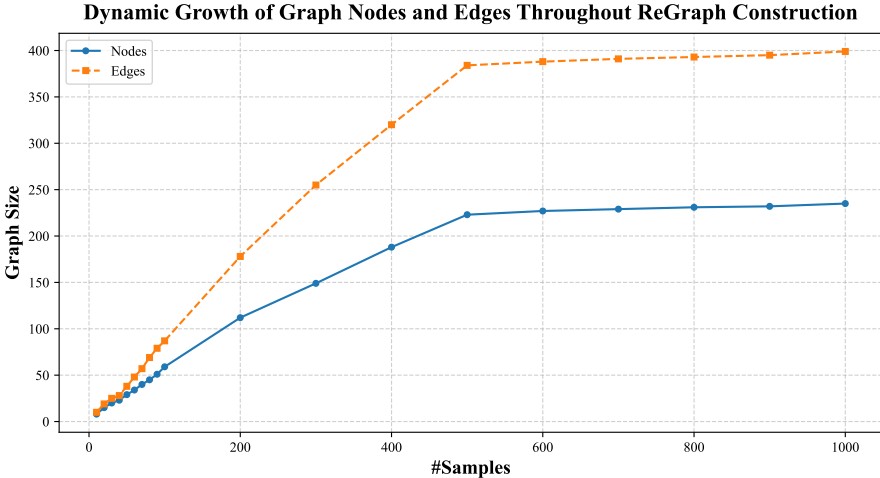

Figure 9: **ReGraph Distribution Across Different Sample Numbers.**

Moreover, we further investigated the impact of ReGraph size on code generation quality by integrating different ReGraphs into Qwen2.5-Coder-7B-Instruct, with 200 search budgets and rollout 10. According to Table 3, we observe that the performance continues to improve with increasing ReGraph size before convergence, demonstrating the effectiveness of the CUDA optimization search space exposed by ReGraph. However, once ReGraph has converged, its size no longer has a significant impact on performance.

Table 3: Ablation study on the impact of ReGraph size on code generation performance.

| ReGraph ID | 0 | 1 | 2 | 3 | 4 | 5 | 6 |
|---|---|---|---|---|---|---|---|
| *Graph Statistics* | | | | | | | |
| #Samples | 30 | 50 | 100 | 300 | 500 | 600 | 1000 |
| #Nodes | 20 | 29 | 59 | 149 | 223 | 227 | 235 |
| #Edges | 25 | 38 | 87 | 225 | 384 | 388 | 399 |
| *Performance Metrics* | | | | | | | |
| pass@10 | 61.3 | 60.1 | 63.9 | 64.9 | 70.0 | 68.4 | 71.2 |
| speedup@10 | 9.43 | 9.74 | 11.29 | 12.63 | 14.15 | 14.09 | 14.21 |

## D    ANALYSIS OF OVERHEAD FOR REGRAPHT

We provide both a theoretical analysis and empirical wall-time measurements for the overhead of the ReGraphT framework, encompassing ReGraph construction as well as the MCGS process. While ReGraph construction incurs LLM-related costs, we break down its three main stages, assuming $N$ samples:

**LLM Reasoning for Trajectories:** Each sample requires a single LLM invocation, resulting in a total of $N$ calls. With a parallelism degree of $C$, the time complexity is $O(N/C)$.

**Relabeling Optimization Methods:** This stage also involves $N$ LLM calls, but they must be executed sequentially, yielding a complexity of $O(N)$.

**Merging into ReGraph:** Each merge operation has complexity $O(M)$, where $M$ denotes the average trajectory length, resulting in an overall complexity of $O(NM)$.

When accounting for parallelism, the dominant cost becomes $O(NM/C)$. Importantly, this represents a one-time overhead, as the ReGraph can be preconstructed and reused. As illustrated in Figure 9, ReGraph typically converges with approximately 500 samples, further enhancing cost efficiency.

In addition to the asymptotical complexity analysis of ReGraph construction, we also provide empirical measurements of the time overhead for ReGraph construction under different sample scales. Figure 10 illustrates the runtime required to construct a ReGraph with 500 samples, as well as the overhead distribution across different components of the construction process. The tests were conducted with a parallelism level of 5, meaning that five parallel threads were used to extract CUDA optimization trajectories. The results show that the time consumption scales approximately linearly with the size of ReGraph. On the other hand, analyzing the overhead distribution across components reveals that the main bottleneck in ReGraph construction lies in verifying the correctness of generated results, which accounts for 48% of the total runtime. In addition, the trajectory generation and relabeling steps, which require model-based generation, contribute 32% and 19% of the total overhead, respectively. In contrast, the cost associated with merging is negligible.

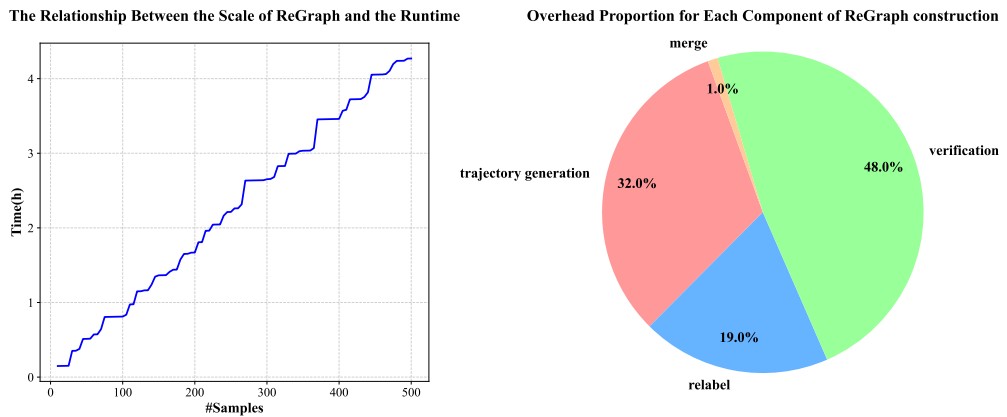

Figure 10: **ReGraph Overhead.**

ReGraphT models optimization as state transitions, making the search space grow linearly with the number of edges rather than nodes. A naive enumeration explores every edge with multiple attempts, yielding a time complexity of $O(CE)$, where $E$ is the number of edges and $C$ is the number of attempts per node. Since a converged ReGraph typically has hundreds of edges, this approach may require thousands of attempts. In contrast, MCGS distributes trials during the rollout phase, avoiding repeated edge attempts. Its time complexity is $O(Nb(d+l))$, where $N$ is the number of iterations, $b$ is branching factor and $d,l$ represents the depth of selection and rollout respectively. Thus MCGS offers more efficient exploration of ReGraph search space.

Furthermore, we provide empirical data on the code generation overhead of ReGraphT. Inference tests were conducted on a single A100-80GB GPU paired with an Intel(R) Xeon(R) Platinum 8358P CPU @ 2.60 GHz, utilizing the Qwen2.5-Coder-7B-Instruct model, with a search budget of 100 and a batch size of 16. Under these conditions, generating results for the 80 CUDAEval samples takes approximately 6.02 hours. Moreover, based on benchmark tests of various sub-8B models running on a single RTX 4090 LLC (2025), comparable results can be achieved on consumer-grade hardware in about 7.53 hours.

# E    ABLATIONS ON MCGS TRAVERSAL

We conduct an ablation study on the varying number of rollouts and different reward strategies to explore the impact of during MCGS traversal. For ReGraphT-MCGS, max attempts is the same as max rollouts. As shown in Table 4, under fixed search budgets and varying configurations, ReGraph-MCGS outperforms traversal-based methods, demonstrating both higher search efficiency and effectiveness. What's more, as the maximum number of rollouts increases, ReGraphT-MCGS demonstrates sustained performance gains.

Table 4: Performance of Different Search Methods under Varying Budgets

| Search Budgets | Search Methods | Max Attempts | Avg. Trajectory Length | pass@n | speedup@n |
|---|---|---|---|---|---|
| 100 | ReGraphT | 5 | 3.1 | 64.9 | 7.91 |
| | | 10 | 2.3 | 63.9 | 7.76 |
| | ReGraphT-MCGS | 5 | 2.5 | 64.9 | 8.13 |
| | | 10 | 2.9 | 64.9 | 8.46 |
| | | 20 | 3.6 | 65.5 | 8.91 |
| 200 | ReGraphT | 5 | 4.9 | 70.0 | 11.62 |
| | | 10 | 4.6 | 68.7 | 11.30 |
| | ReGraphT-MCGS | 5 | 5.2 | 66.1 | 11.99 |
| | | 10 | 5.5 | 70.0 | 12.88 |
| | | 20 | 6.3 | 71.2 | 13.43 |
| 300 | ReGraphT | 5 | 7.4 | 68.7 | 12.89 |
| | | 10 | 6.8 | 70.0 | 12.45 |
| | ReGraphT-MCGS | 5 | 7.8 | 69.6 | 13.01 |
| | | 10 | 8.7 | 72.5 | 14.76 |
| | | 20 | 9.1 | 74.1 | 14.98 |

To investigate the effect of reward formulation, we considered three different reward strategies:

**strict reward** The reward is defined as the average performance speedup of designs that pass all unit tests:

$$R_{\text{strict}} = \frac{1}{|\mathcal{D}_{\text{pass-all}}|} \sum_{d \in \mathcal{D}} p(d) \cdot \mathbb{1}\left[\bigwedge_{t \in \mathcal{T}} \text{pass}(d, t)\right],$$

where $\mathcal{D}_{\text{pass-all}}$ denotes the subset of designs that successfully pass every test in $\mathcal{T}$.

**partial-credit reward** Compared to strict reward, partial-credit reward does not require all unit tests to be passed. Instead, it allocates rewards proportionally to the fraction of unit tests that are successfully passed:

$$R_{\text{partial}} = \frac{1}{m} \sum_{t \in \mathcal{T}} \frac{1}{|\mathcal{D}_t|} \sum_{d \in \mathcal{D}} p(d) \cdot \mathbb{1}[\text{pass}(d, t)],$$

where $\mathcal{D}_t$ is the set of designs passing test $t$.

**rollout-based reward** Similar to Baronio et al. (2025), rollout-based reward models the reward as a Markov decision process (MDP), setting the reward of a given response as the discounted sum of scores of the current kernel and all subsequent ones and provides fine-grained feedback during generation:

$$R_{\text{rollout}} = \mathbb{E}\left[\sum_{t=0}^{T} \gamma^t r(s_t, a_t)\right],$$

where $r(s_t, a_t)$ denotes the performance gain associated with the design choice at step $t$, and $\gamma \in [0, 1]$ is the discount factor.

Under different reward strategies, We conduct experiments using Qwen2.5-Coder-7B-Instruct as the SLM and a ReGraph built from 500 samples in Figure 9. The search budgets and varying rollout configurations are fixed to 200 and 10. As shown in Figure 11, we observe that strict reward and rollout-based reward have similar performance, while partial-credit reward leads to a slightly lower pass rate and speedup performance.

# F   LIMITATIONS

During the construction of CUDAEval, we employed a multi-stage pipeline to carefully filter and select high-quality CUDA samples. While this process ensured the reliability and consistency of the benchmark, it resulted in the exclusion of a substantial portion of the original dataset. Consequently, CUDAEval may not fully capture the diversity and complexity of real-world CUDA

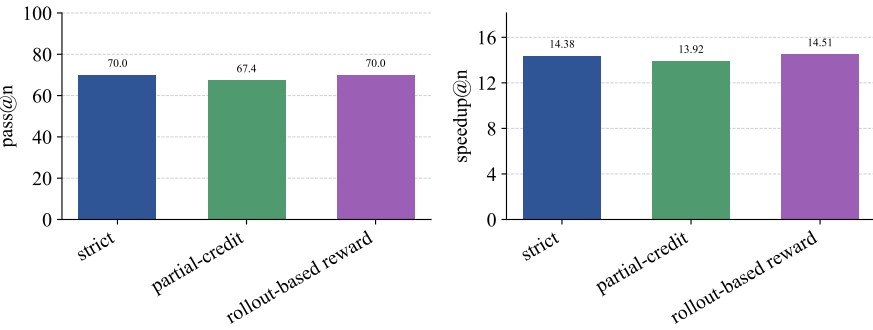

Figure 11: **Ablation Study on Different Reward Strategies.**

programs, particularly those with uncommon patterns, intricate dependency structures, or unconventional optimization strategies. This selective filtering could limit the evaluation of models on edge cases or rare optimization scenarios, potentially underrepresenting certain challenging aspects of CUDA code generation and optimization. Future work may focus on incorporating a broader variety of samples to create a more comprehensive and representative benchmark.

## G PROMPTS

### G.1 PROMPT FOR KERNEL EXTRACTION

**(a) Prompt for CUDA Kernel Extraction**

```
**CUDA kernel process prompt**

**Role**:
You are a professional high performance computing(HPC) engineer,
skilled in optimizing C++ serial code using CUDA.

**Responsibility**:
You are supposed to extract the CUDA kernels from the given CUDA code
file and identify the optimization techniques used in them.
If the provided CUDA code file contains multiple CUDA kernels, you
should extract all of them and for each of them analyze all
optimizations used and corresponding code snippet.

**Response Format**:
```json
{
    "kernels": [

        {
            "name": <extracted cuda kernel name>,
            "content": <extracted cuda kernel content>
        }
    ],
    "optimizations": [
        [
            {
                "optimization": <the optimization method used>,
                "snippet": <corresponding code snippet>
            },
        ]
    ]
```

```
}
```

**Precautions**
1. You must only return the kernels that exist within this file, not
those imported from other files and merely called here.
2. For each kernel, you must include its complete content without any
omissions or abbreviated formatting.
3. Ensure that in the returned JSON content, the length of kernels
matches the length of optimizations, meaning each kernel corresponds
to a list of optimizations.
```

## G.2 PROMPT FOR DEPENDENCY

### (a) Prompt for CUDA Kernel Dependency Completion

```
You are an HPC engineer proficient in using CUDA. The CUDA kernel is
extracted from the code file, so it may lack some relevant
dependencies.
Now for the CUDA kernel provided by the user, you need to determine
whether this CUDA kernel lacks relevant dependencies.
1. If it lacks standard library dependencies, please supplement them.
2. If it lacks user file dependencies, for example, user-defined
classes, user-defined functions, user-defined macros, etc., attempt to
rewrite it in a simple manner to resolve the dependency issues.

Please return whether the rewrite was successful. If the rewrite is
successful, return the rewritten code. If you are unable to rewrite
the required user dependencies, return None for this item.

Note: that the user's code where this kernel resides is unavailable.
Therefore, if you think some definitions are likely defined in
the user's code, you are also supposed to attempt to supplement them
as part of the rewritten code.

# Prompt format

The user will provide you a JSON dictionary in the following format:

```json
{
    "kernel" : <The CUDA kernel provided by user>
}
```

# Response format

You will respond with a JSON dictionary in the following format:

```json
{
    "success": "<yes/no>",
    "reason": "<Your reasoning process>",
    "rewrite: "<The rewritten code that doesn't lack relevant
dependencies/None>"
}
```
```

## G.3 PROMPT FOR CUDA REASONING

### (a) Prompt for CUDA Optimization Reasoning

```
You are an excellent high-performance computing engineer,
skilled in optimizing CPP code using CUDA.
Now, the user will provide you with CPP code,
and you need to optimize it step by step using CUDA.

# Notes
1. Please optimize CUDA step by step. In each step of the optimization
process, you need to provide the reasoning behind the optimization,
explain the optimization methods used, and describe how these methods
are applied. Finally, provide the optimized code. Optimization methods
refer to CUDA optimization techniques such as shared memory, warp
divergence elimination etc. 'How the optimization methods are used'
refers to how these CUDA optimization techniques are applied to
optimize the code.
2. The optimization process should be returned as a JSON list.
3. The function name must remain the same as the initial function
after each optimization step.

# Prompt Format

The user will provide a JSON dictionary in the following format:

```json
{
    "kernel": "<The CPP code provided by user>",
}
```

# Response Format

You should respond in the following JSON format:

```json
[
    {
        "think": "<The thought process for this optimization step>",
        "method": "<The optimization method used>",
        "detail": "<How the optimization methods are used>",
        "code": "<The optimized code obtained in this step>"
    }
]
```
```

## G.4 PROMPT FOR RELABEL

### (a) Prompt for CUDA Optimization Relabel

```
You are an excellent high-performance computing engineer, skilled in
optimizing CPP code using CUDA. Now, the user will provide you with a
step-by-step optimization process for CPP code along with some
existing CUDA optimization methods. You need to determine whether each
CUDA optimization method used in this step-by-step process falls
within the scope of the existing CUDA optimization methods.

If the method used is part of the existing methods, rename it to the
corresponding method name from the existing ones; otherwise, keep the
optimization method's name unchanged.
```

```
# Notes
1. The user input is a json dict incluing 2 lists, 'methods'
represents the existing CUDA optimization methods, and 'process'
represents the optimization process, where each item represents one
optimization step.
2. For each optimization step, you need to make a judgment.
3. The CUDA optimization method used in each step is indicated in the
'method' field.
4. You should return a list in JSON format, with the same length as
the input list.

# Prompt Format

The user will provide a JSON dictionary in the following format:

```json
{
    "methods: [<CUDA optimization methods existed>],
    "process": [
        {
            "think": "<The thought process for this optimization
            step>",
            "method": "<The optimization method used>",
            "detail": "<How the optimization methods are used>",
            "code": "<The optimized code obtained in this step>"
        }
    ]
}
```

# Response Format

You should respond in the following JSON format:

```json
[
    {
        "existed": "<yes/no>",
        "method": "<If yes, the corresponding method name from the
        existing methods; if no, keep the original method name>"
    }
]
```
```

## G.5 PROMPT FOR STANDARD

### (a) Prompt for Standard

```
You are an excellent high-performance computing engineer, skilled in
optimizing CPP code using CUDA. Now, the user will provide you with
CPP code, and you need to optimize it using CUDA.

# Notes
1. You need to use CUDA to optimize the CPP code provided by user.
2. The optimized function name needs to remain consistent with the
original function. You need to handle the data transfer between host
(CPU) memory and device (GPU) memory, as well as the invocation of
CUDA kernels, within the function.
3. You must provide the complete code without any omissions.
```

```
# Prompt Format

The user will provide a JSON dictionary in the following format:

```json
{
    "kernel": "<The CPP code provided by user>",
}
```

# Response Format

You should respond in the following JSON format:

```json
{
        "think": "<The thought process for this optimization>",
        "code": "<The optimized code using CUDA>"
}
```
```

## G.6 PROMPT FOR COT

### (a) Prompt for CoT

```
You are an excellent high-performance computing engineer, skilled in
optimizing CPP code using CUDA. Now, the user will provide you with
CPP code, and you need to optimize it step by step using CUDA.

# Notes
1. Please optimize CUDA step by step. In each step of the optimization
process, you need to provide the reasoning behind the optimization,
explain the optimization methods used, and describe how these methods
are applied. Finally, provide the optimized code. Optimization methods
refer to CUDA optimization techniques such as shared memory, warp
divergence elimination etc. 'How the optimization methods are used'
refers to how these CUDA optimization
techniques are applied to optimize the code.
2. The optimization process should be returned as a JSON list.
3. The function name must remain the same as the initial function
after each optimization step. You need
to handle the data transfer between host (CPU) memory and device (GPU)
memory, as well as the invocation of CUDA kernels, within the function.
4. You must provide the complete code without any omissions.

# Prompt Format

The user will provide a JSON dictionary in the following format:

```json
{
    "kernel": "<The CPP code provided by user>",
}
```

# Response Format

You should respond in the following JSON format:

```json
[
```
```

```
    {
        "think": "<The thought process for this optimization step>",
        "method": "<The optimization method used>",
        "detail": "<How the optimization methods are used>",
        "code": "<The optimized code obtained in this step>"
    }
]
```
```

## G.7 PROMPT FOR CODERAG

**(a) Prompt for CodeRAG**

```
You are a coding expert that writes very fast code. You write parallel
C and C++ code using CUDA and always strive to make the code as fast
as possible. The user will give you code and you will provide a
modified version of the user's code that is as fast as possible using
CUDA. At the same time, the user will also provide an optimization
example, including the original program and the optimized program
using CUDA. You can refer to this optimization example for your own
optimization.

# Prompt format

The user will provide you a JSON dictionary in the following format:

```json
{
    "source_code" : <Initial code>,
    "example_orginal" : <Example original program>,
    "example_optimized": <Example optimized program>
}
```

# Response format

You will respond with a JSON dictionary in the following format:

```json
{
    "updated_code" : <Optimized code>
}
```

"""
```

## G.8 PROMPT FOR REGRAPHT

**(a) Prompt for ReGraphT**

```
You are a coding expert that writes very fast code. You write parallel
C and C++ code using CUDA and always strive to make the code as fast
as possible. The user will give you code and you will provide a
modified version of the user's code that is as fast as possible using
CUDA.
At the same time, the user will also provide an optimization example,
```

```
including an optimization example consisted of the original program
and the optimized program using CUDA, and the CUDA optimization method
used.
This optimization example may not necessarily apply to the current
code to be optimized, so you also need to determine whether the
provided optimization method
is suitable.

# Prompt format

The user will provide you a JSON dictionary in the following format:

```json
{
    "source_code" : <Initial code>,
    "example": {
        "origin": <The original program in the optimization example>,
        "optimized": <The optimized program using CUDA in the
        optimization example>,
        "method": <The CUDA optimization method used in the
        optimization example>
    },
}
```

# Response format

You will respond with a JSON dictionary in the following format:

```json
{
    "suitable": <If the provided optimization method is suitable,
    yes/no>,
    "optimization": <The optimized code using CUDA>
}
```
```

## H  LLM USAGE

In preparing this manuscript, we employed a large language model (LLM) solely for grammar correction and stylistic polishing of the text. The LLM was not used for developing research ideas, designing methodologies, conducting experiments, or analyzing results. All scientific contributions, including problem formulation, theoretical analysis, experimental design, implementation, and evaluation, were carried out entirely by the authors.

