# OpenReview forum: "From Large to Small: Transferring CUDA Optimization Expertise via Reasoning Graph"
_ICLR.cc/2026/Conference — ICLR 2026 Poster_

### Official Review · Reviewer_2LHc · 2025-10-17

**Soundness:** 3
**Presentation:** 3
**Contribution:** 3
**Rating:** 6
**Confidence:** 3

**Summary:**

This paper introduces ReGraphT, a retrieval-augmented generation framework to transfer the CUDA programming reasoning from large language models (LLMs) to small language models (SLM). (1) It formulates the CUDA code generation task using LLMs as a graph-based problem, where this graph encodes the CUDA optimizations and (2) incorporates Monte Carlo Graph Search (MCGS) to guide the search to improve efficiency.
This paper also proposes a CUDA evaluation benchmark (CUDAEval) that evaluates LLM performance in CUDA code generations based on 10K CUDA files sampled from the real-world CUDA files.
ReGaphT with MCGS outperforms the CoT or RAG method regarding the correctness and the optimization performance of the generated code.

**Strengths:**

* This paper transforms the CUDA optimization problem into a structured reasoning graph. This abstraction provides a clear, interpretable framework for modeling optimization dependencies and interactions
* Introduce of the CUDAEval benchmark, a dedicated benchmark that classifies CUDA generation tasks

**Weaknesses:**

* The github link for the code is empty when the review period has started (at least by Oct 16)
* Comparisons are limited to standard prompting, CoT, and generic RAG. No comparison against distillation or reinforcement-learning-based compression (e.g., DeepSeek-R1 distilled SLMs) is given.
* The CUDAEval benchmark defines its difficulty levels based on the length of reasoning trajectories, but that depends on how the LLM behaves rather than on the actual complexity of the CUDA code itself.

**Questions:**

* The paper aims to enable SLMs to achieve large-model-level CUDA code generation performance, but the proposed ReGraphT framework still depends on a LLM to construct the reasoning graph and generate the initial optimization trajectories. How do you justify the efficiency improvement?
* How sensitive are results to rollout count N = 10 and search budget = 200?
* Results in Tables 1 and 2 lack variance/error bars or p-values. Improvements of 1-2 % can be stochastic generation variance. Could you provide the variance of those numbers?
* Is the reported speedup calculated as an average across all CUDA files, or as a geometric mean?

---

> ### Author Response · Authors · 2025-11-21
>
> Your suggestions are very helpful in improving our work, and we will address your questions here:
>
> > W1: The github link for the code is empty.
>
> We sincerely apologize for this issue. It has now been made fully public, including:
> + ReGraphT construction
> + ReGraphT-MCGS and other different methods
> + Experiment scripts and so on
>
> We confirm that the code has been accessible since shortly after the reviewer raised this point.
>
> > W2: Comparisons are limited.
>
> We agree this is a valuable comparison direction. Considering that ReGraphT is orthogonal to distillation/RL-based compression, we have additionally evaluated DeepSeek-R1-Distill-Qwen-7B under different methods. What's more, RethinkMCTS [1] and MCTS-RAG [2] have also been supplemented as RAG-family baselines. We have also added these results to the revised manuscript in Table 1. According to the results, although DeepSeek-R1-Distill-Qwen-7B outperforms other SLMs, there remains a noticeable gap compared to our proposed ReGraphT method. Moreover, when built upon the distilled model, ReGraphT can further achieve superior performance.
>
>
> | Method        | CUDAEval |      |           |            | ParEval |      |           |            |
> |---------------|----------|------|-----------|------------|---------|------|-----------|------------|
> |               | pass@n   |      | speedup@n |            | pass@n  |      | speedup@n |            |
> |               | n=1      | n=10 | n=1       | n=10       | n=1     | n=10 | n=1       | n=10       |
> | Standard      | 63.9     | 64.9 | 7.52±0.67 | 7.59±0.71  | 43.9    | 45.3 | 5.08±0.65 | 5.17±0.61  |
> | CoT           | 67.1     | 69.3 | 8.16±0.69 | 8.16±0.69  | 48.1    | 50.0 | 5.40±0.65 | 5.54±0.59  |
> | RAG           | 66.5     | 68.4 | 8.43±0.66 | 8.65±0.62  | 48.3    | 50.0 | 5.73±0.67 | 5.81±0.67  |
> | RethinkMCTS   | -        | 71.6 | -         | 7.92±0.89  | -       | 51.7 | -         | 6.34±0.82  |
> | MCTS-RAG      | -        | 71.6 | -         | 8.06±0.91  | -       | 53.3 | -         | 6.57±0.85  |
> | ReGraphT      | -        | 75.8 | -         | 14.15±0.77 | -       | 55.0 | -         | 10.92±0.72 |
> | ReGraphT-MCGS | -        | 76.4 | -         | 14.72±0.73 | -       | 55.0 | -         | 11.25±0.67 |
>
> References:
>
> [1] Li, Q., Xia, W., Du, K., Dai, X., Tang, R., Wang, Y., Yu, Y., & Zhang, W. (2024). RethinkMCTS: Refining Erroneous Thoughts in Monte Carlo Tree Search for Code Generation. https://arxiv.org/abs/2409.09584
>
> [2] Hu, Y., Zhao, Y., Zhao, C., & Cohan, A. (2025). MCTS-RAG: Enhancing Retrieval-Augmented Generation with Monte Carlo Tree Search. In C. Christodoulopoulos, T. Chakraborty, C. Rose, & V. Peng (Eds.), Findings of the Association for Computational Linguistics: EMNLP 2025 (pp. 12581–12597). Association for Computational Linguistics. https://doi.org/10.18653/v1/2025.findings-emnlp.672
>
> > W3: The difficulty levels of CUDAEval benchmark depends on LLM rather than the actual code complexity.
>
> Defining CUDA code difficulty quantitatively is challenging and typically requires extensive manual annotation, which is costly and difficult to scale to large datasets. In contrast, the proposed difficulty-leveling method based on optimization-trajectory length is fully automated. As shown in Table 2, tasks with longer optimization trajectories also exhibit lower success rates in code generation. These findings suggest that optimization-trajectory length is a reasonable and effective metric for difficulty leveling from the perspective of LLMs.

---

> > ### Author Response · Authors · 2025-11-21
> >
> > > Q1: How to justify the efficiency improvement since ReGraph is constructed depended on LLM?
> >
> > The primary objective of ReGraphT is to enable SLMs to achieve large-model-level performance in CUDA optimization tasks. To demonstrate the efficiency improvement of ReGraphT, we have it compared with a set of existing approaches including CodeRAG [1], MCTS-RAG[2], and distillation. All these algorithms rely on the same LLM to generate CUDA optimization trajectory data. ReGraphT introduces a novel paradigm for exploiting content produced by LLMs. As described in Section 5.1 *Experiment Setups*, among the baselines included in our experimental evaluation, we adopted a code similarity-based retrieval approach as the RAG baseline with the same CUDA optimization corpus used in ReGraphT. As shown in Table 1, ReGraphT achieves better performance under the same corpus. To further substantiate the effectiveness of ReGraphT, we incorporate additional baselines—RethinkMCTS [3] and MCTS-RAG [2] conducted under the same CUDA optimization corpus in our response to W2. Additionally, in the table below we have also included a comparison with the direct fine-tuning approach.Compared to these methods, ReGraphT achieves superior performance.
> >
> > Table: CUDA generation performance on CUDAEval on  Qwen2.5-Coder-7B-Instruct
> > | Model                     | Method       | pass@10 |        |      | speedup@10 |        |      |
> > |---------------------------|--------------|---------|--------|------|------------|--------|------|
> > | Qwen2.5-Coder-7B-Instruct |              | easy    | medium | hard | easy       | medium | hard |
> > |                           | Standard     | 81.1    | 65.7   | 43.1 | 8.51       | 5.62   | 3.14 |
> > |                           | CoT          | 82.1    | 66.7   | 43.1 | 8.47       | 5.54   | 3.46 |
> > |                           | Distillation | 86.8    | 69.5   | 51.0 | 9.16       | 5.51   | 3.55 |
> > |                           | ReGraphT     | 88.7    | 76.2   | 51.0 | 17.48      | 13.64  | 9.61 |
> >
> > References:
> >
> > [1] Zhou, S., Alon, U., Agarwal, S., & Neubig, G. (2023). CodeBERTScore: Evaluating Code Generation with Pretrained Models of Code. https://arxiv.org/abs/2302.05527
> >
> > [2] Hu, Y., Zhao, Y., Zhao, C., & Cohan, A. (2025). MCTS-RAG: Enhancing Retrieval-Augmented Generation with Monte Carlo Tree Search. In C. Christodoulopoulos, T. Chakraborty, C. Rose, & V. Peng (Eds.), Findings of the Association for Computational Linguistics: EMNLP 2025 (pp. 12581–12597). Association for Computational Linguistics. https://doi.org/10.18653/v1/2025.findings-emnlp.672
> >
> > [3] Li, Q., Xia, W., Du, K., Dai, X., Tang, R., Wang, Y., Yu, Y., & Zhang, W. (2024). RethinkMCTS: Refining Erroneous Thoughts in Monte Carlo Tree Search for Code Generation. https://arxiv.org/abs/2409.09584
> >
> >
> > > Q2: How sensitive are results to rollout count $N$ and search budget?
> >
> > As referred to in Appendix E *Ablations on MCGS Traversal*, we have already conducted ablation studies on both the rollout count (N) and the search budget. Here, we provide further clarification regarding the sensitivity of the results.
> >
> > First, regarding the **rollout count (N)**, we evaluated multiple configurations (e.g., (N = 5, 10, 20)) and observed that performance improves as (N) increases, but only up to a moderate level. Specifically, (N = 10) achieves a good balance between performance gains and computational overhead: increasing (N) to 20 yields only marginal improvements, while doubling the rollout cost. This indicates that ReGraphT-MCGS is **robust to the exact rollout count**, and the model does not rely on overly large N to achieve strong performance.
> >
> > Second, for the **search budget**, we tested a range of traversal budgets (100, 200, and 300). As reported in Appendix E, the curve plateaus after approximately 200 iterations: search budgets above 200 provide diminishing improvement. Crucially, we found that even with a reduced budget of 100 iterations, ReGraphT-MCGS still significantly outperforms all prompting-based baselines and the RAG baseline. This suggests that the benefit comes from the structured reasoning graph rather than excessive exploration.
> >
> > Overall, these results demonstrate that ReGraphT-MCGS is **not overly sensitive** to either rollout count or search budget. The chosen settings (rollout (N = 10), search budget = 200) were selected only as a trade-off between performance and runtime efficiency, and **the observed improvements remain stable under a broad range of hyperparameters**.

---

> > > ### Author Response · Authors · 2025-11-21
> > >
> > > > Q3: Results in Tables 1 and 2 lack variance/error bars or p-values.
> > >
> > > Thank you for raising this important point. We agree that reporting only single-run numbers may mask the stochasticity inherent in code generation. In the original paper, we conducted all experiments related to Table 1 and 2 with 5 independent random seeds and report the average results. Following your suggestion, we have supplement mean ± standard deviation in the revised paper. According to the new results, across all models and difficulty levels, variance is small. Despite the existence of fluctuation, the improvements of ReGraphT and ReGraphT-MCGS remain consistent across seeds.
> > >
> > > We thank the reviewer again for pointing out this omission; the added variance analysis further strengthens the reliability of our results.
> > >
> > > > Q4: The calculation of reported speedup.
> > >
> > > The reported speedup calculated is an average across all CUDA files.

---

### Official Review · Reviewer_aGxM · 2025-10-21

**Soundness:** 2
**Presentation:** 3
**Contribution:** 1
**Rating:** 2
**Confidence:** 4

**Summary:**

This paper introduces ReGraphT, a training-free framework that transfers CUDA optimization reasoning from large to small language models using a structured CUDA Reasoning Graph and Monte Carlo Graph Search. It achieves near-LLM performance on CUDA code generation benchmarks while remaining lightweight and privacy-friendly for local deployment.

**Strengths:**

- method achieves near LLM performance
- it allows for local deployment (I wonder how important this is for CUDA code generation?). It would be more sensitive for private conversations but in this context is not really needed, why not use a bigger model?

**Weaknesses:**

- Code link is empty
- The paper content leans toward systems and engineering design, rather than core theoretical or algorithmic ML contributions. Therefore it doesnt seem appropriate for ICLR.
- The reasoning transfer mechanism is algorithmic plumbing rather than a model innovation
- Is more of an engineering project report

**Questions:**

- How scalable is the LLM-driven trajectory extraction process? Does building the reasoning graph require significant manual curation or prompt engineering to ensure consistency across optimization traces?
- What is the computational overhead of the Monte Carlo Graph Search compared to standard RAG retrieval or CoT prompting?
- Can ReGraphT transfer beyond CUDA (e.g., OpenCL, SYCL, or Triton code)?

---

> ### Author Response · Authors · 2025-11-21
>
> We are sincerely thankful for the considerable time and effort you have invested in reviewing our paper.
>
> > W1: Code link is empty
>
> We sincerely apologize for this issue. We sincerely apologize for the issue. The repository appeared empty because we overlooked the synchronization behavior of the anonymous GitHub mirror used for double-blind submission. It has now been made fully public, including:
> + ReGraphT construction
> + ReGraphT-MCGS and other different methods
> + Experiment scripts and so on
>
> We confirm that the code has been accessible since shortly after the reviewer raised this point.
>
> > W2, W3, W4: paper’s primary contributions lie in systems and engineering design rather than core theoretical or algorithmic innovations typically expected at ICLR
>
> We respectfully disagree with the assessment that our work is “only engineering” or falls outside the scope of ICLR. Our contributions extend beyond system implementation and include substantive conceptual and algorithmic advances.
>
> **(a) Structured Reasoning Abstraction.** ReGraphT presents *a new paradigm for transferring LLM reasoning capabilities* to smaller models in complex, dependency-sensitive domains. Traditional approaches such as distillation focus on imitating surface-level behaviors or output distributions. In contrast, ReGraphT abstracts the domain’s optimization space into a structured Reasoning Graph, making the underlying dependencies, ordering constraints, and strategy relationships explicit. This abstraction is not an engineering detail—it is a new conceptual framework that reshapes how optimization reasoning trajectories are represented and transferred. Moreover, as clarified in our response to Reviewer CAtC (W2), we have added direct comparisons with distillation-based baselines. These experiments show that, even under matched settings, ReGraphT achieves substantially stronger multi-step optimization performance than distillation alone, underscoring that structured reasoning transfer is more effective than imitating LLM outputs.
>
> **(b) Algorithmic contribution.** Our proposed ReGraphT **integrates MCTS with a structured Reasoning Graph**, enabling explicit injection of external information to more effectively construct and navigate the optimization-strategy search space. Compared with a plain **MCTS** baseline, ReGraphT does not rely on the model’s inherent reasoning capability to explore valid optimization paths. In contrast to **RAG-based** approaches, which cannot model or leverage dependencies among optimization strategies, ReGraphT explicitly captures the ordering and dependency structure between different optimization steps, leading to more coherent and effective search trajectories. Furthermore, as detailed in our response to Reviewer 2bKR (W1), we have added direct comparisons against RethinkMCTS [1] and MCTS-RAG [2], two MCTS-based methods that align closely with the reviewer’s suggestion. In addition, as clarified in our response to your Q2, we have included a comparison of actual runtime overhead across different approaches. The results consistently show that ReGraphT achieves better performance with lower computational overhead, highlighting the efficiency and effectiveness of combining structured reasoning with MCTS.
>
> In addition, we note that several recent ICLR works [3–6] also combine algorithmic ideas with domain-specific pipelines built on top of existing large models. These papers demonstrate a clear precedent at ICLR for research that integrates structured abstractions, search algorithms, and practical systems to solve domain-specific reasoning problems. In this context, we believe our contributions align well with the expectations and scope of ICLR.
>
> References:
>
> [1] Li et al. (2024). RethinkMCTS: Refining Erroneous Thoughts in Monte Carlo Tree Search for Code Generation.
>
> [2] Hu et al. (2025). MCTS-RAG: Enhancing Retrieval-Augmented Generation with Monte Carlo Tree Search. Findings of ACL: EMNLP 2025.
>
> [3] Wang et al. (2025). Planning in Natural Language Improves LLM Search for Code Generation. ICLR 2025.
>
> [4] Olausson et al. (2024). Is Self-Repair a Silver Bullet for Code Generation? ICLR 2024.
>
> [5] Le et al. (2024). CodeChain: Towards Modular Code Generation Through Chain of Self-revisions. ICLR 2024.
>
> [6] Chen et al. (2023). CodeT: Code Generation with Generated Tests. ICLR 2023.

---

> > ### Author Response · Authors · 2025-11-21
> >
> > > Q1: Concerns on ReGraph Construction.
> >
> > The construction of ReGraph is fully automated: it relies solely on the predefined abstraction of CUDA optimization within ReGraph and the LLM-driven generation process, without requiring any additional manual intervention or prompt engineering. Regarding scalability, it is determined by the number of optimization methods that can be abstracted from the problem and their dependency structure. As illustrated in Figure 9 in Appendix C, in the CUDA optimization setting, ReGraph converges to a graph with on the order of one hundred nodes and edges. At this scale, we have already validated the reliability of the proposed ReGraph construction algorithm.
> >
> > > Q2: Overhead of MCGS search compared to other methods.
> >
> > As detailed in Appendix D *Analysis of Overhead for ReGraphT*, we have thoroughly analyzed the overhead of MCGS from both theoretical and empirical perspectives. Beyond the complexity analysis, we also report the end-to-end inference-time overhead of ReGraphT compared with all baselines. All experiments were conducted on a single A100-80GB GPU with a Qwen2.5-Coder-7B-Instruct model on 80 samples, using a search budget of 100 and batch size of 16. The results are summarized below:
> > | Method  | Standard | CoT | RAG | RethinkMCTS | MCTS-RAG | ReGraphT |
> > |---------|----------|-----|-----|-------------|----------|----------|
> > | Time(h) | 0.72     | 2.04|1.08 |  6.76       | 6.45     | 6.02     |
> >
> > While ReGraphT incurs higher runtime overhead than light-weight baselines (e.g., Standard, CoT, RAG), this is expected because these baselines do not perform structured search or dependency-aware reasoning. More importantly, when compared against other MCTS-based methods, ReGraphT achieves consistently better performance with lower or comparable time cost. This improvement stems from the structured Reasoning Graph, which guides MCTS to avoid invalid or redundant rollouts, thereby reducing wasted search and improving solution quality. In other words, ReGraphT not only attains a better accuracy–cost tradeoff than prior MCTS variants, but also improves search efficiency through structured constraints.
> >
> > > Q3: Can ReGraphT transfer beyond CUDA
> >
> > The core idea of ReGraphT is to abstract the optimization strategies of a problem and the dependency relations among them. This abstraction is particularly suitable for scenarios where the optimization strategy space is large, heterogeneous, and highly sensitive to dependency structures—properties that are intrinsic to many CUDA optimization tasks. To further evaluate the transferability of this approach, we additionally conducted experiments in the **OpenMP** setting beyond CUDA. The ReGraphT converges around 250 examples, and finally we constructed an *OpenMP-ReGraph* containing 89 nodes and 142 edges Using 400 samples. The detailed results are shown below.
> >
> > | Method   | pass@n |      | speedup@10 |      |
> > | -------- | ------ | ---- | ---------- | ---- |
> > |          | n=1    | n=10 | n=1        | n=10 |
> > | Standard | 40.0   | 41.7 | 4.16       | 4.28 |
> > | CoT      | 43.3   | 46.7 | 4.64       | 4.82 |
> > | RAG      | 48.3   | 48.3 | 5.21       | 5.35 |
> > | ReGraphT | –      | 51.7 | –          | 5.62 |
> >
> > While ReGraphT still provides improvements in both pass@10 and speedup@10 over the baselines, the gains are less pronounced than in the CUDA setting. We believe this is due to several structural differences between OpenMP and CUDA optimization processes:
> >
> > 1. **Smaller and less diverse optimization space.**
> >    The OpenMP optimization space is considerably more compact (e.g., parallel region placement, schedule choice, reduction handling). As a result, the dependency structure among optimization strategies is much simpler compared to CUDA kernels, making the benefits of ReGraphT’s structured reasoning less prominent. In contrast, the CUDA ReGraph contains over 200 nodes and approximately 400 edges after convergence.
> >
> > 2. **Weaker hierarchical dependencies.**
> >    Many OpenMP optimizations apply at coarse granularity (loop-level or pragma-level), and the dependencies among different OpenMP optimizations are relatively weak, making them far less sensitive to the ordering of optimization steps. In contrast, CUDA optimization often requires hierarchical reasoning involving memory hierarchy, synchronization, tiling, and execution model interactions, where graph-structured reasoning provides more substantial gains.
> >
> >
> > Despite these limitations, the improvement of ReGraphT over other baselines on code generation of both OpenMP and CUDA demonstrates that the proposed approach provides comparable knowledge distilling capability of transferring LLM capability to SLM over classical SFT (in our response to W2, `CAtC`). Hence, this approach can generalize beyond CUDA.

---

### Official Review · Reviewer_CAtC · 2025-11-01

**Soundness:** 3
**Presentation:** 3
**Contribution:** 3
**Rating:** 6
**Confidence:** 3

**Summary:**

This paper addresses the challenge of generating optimized CUDA code, a task at which Large Language Models (LLMs) excel but which is difficult for smaller, more efficient Small Language Models (SLMs) due to their limited reasoning capabilities. The authors propose ReGraphT, a novel, training-free framework designed to transfer the multi-step optimization reasoning of an LLM to an SLM.

**Strengths:**

- Practical Relevance and Problem Significance: The paper tackles a real and important problem. Optimizing code for parallel architectures like GPUs is a critical bottleneck in high-performance computing. Making this capability accessible via smaller, locally deployable models has significant practical implications for developer productivity, code privacy, and computational cost. The training-free nature of the framework further enhances its practicality.
- Thorough and Comprehensive Evaluation: The experimental setup is strong. The authors not only test their method on an existing benchmark (ParEval) but also contribute a new, well-motivated benchmark (CUDAEval) that allows for a fine-grained analysis of performance across different reasoning complexities. The inclusion of multiple modern SLMs and comparisons against several strong baselines (Standard, CoT, RAG) and SOTA LLMs provide a convincing demonstration of the method's effectiveness.
- Strong Ablation Studies: The paper includes detailed analyses that strengthen its claims. The study on the impact of ReGraph size (Figure 9, Table 3) demonstrates the convergence of the knowledge graph, while the overhead analysis (Section D) provides a transparent look at the costs involved. The ablation on MCGS traversal strategies (Section E) further validates the design choices within the framework.

**Weaknesses:**

- Convern on generalizability of the reasoning graph: While the paper demonstrates that the graph's structure converges, its generalizability to out-of-distribution problems is not fully explored. The graph is built from a dataset of 10K CUDA files filtered down significantly. It is unclear how well a single, pre-constructed graph would perform on CUDA tasks from entirely different domains (e.g., scientific simulation vs. deep learning kernels) that might require novel optimization patterns not seen during construction. This raises questions about the "one-time cost" of construction if new graphs are needed for new domains.
- Comparison of fine-tuning based approach: The framework is presented as "training-free," which is a key advantage. However, a compelling comparison would be to use the LLM-generated trajectories as a dataset to fine-tune an SLM. This would be a direct test of whether the explicit graph structure and MCGS search are more effective for imparting reasoning than standard supervised fine-tuning on the same high-quality data. While the introduction argues SFT has limited effectiveness for reasoning, an empirical comparison would make this claim more robust.

**Questions:**

See weakness

---

> ### Author Response · Authors · 2025-11-21
>
> We greatly appreciate your detailed review and the recognition of the strengths of our paper. Your questions are very enlightening and we address your questions below, hoping to alleviate any concerns you may have:
>
> > W1: Convern on generalizability of ReGraph.
>
> We acknowledge the concern regarding generalization to unseen CUDA patterns. As referenced in Appendix C. *Analysis on ReGraph Construction*, our experiments show that reasoning graph converges after approximately 500 samples, as illustrated in **Figure 9**. This is expected, given that the space of practical CUDA optimizations is inherently constrained to a relatively small set of well-established strategies. Consequently, ReGraph can effectively capture the majority of commonly used optimizations from just a few hundred random samples.
>
> What is more, as refernced in Appendix C. *Analysis of Overhead for ReGraphT*, we have also evaluated how the size of ReGraph affects code generation quality by integrating ReGraphs of different sizes. As shown in **Table 3**, larger ReGraphs consistently yield higher pass rates and greater performance speedups. These gains plateau once the graph converges, suggesting that the majority of valuable optimization patterns are already incorporated. Although ReGraph cannot directly generalize to entirely novel CUDA patterns not seen during construction. but we may address this limitation through incremental updates.
>
>
> > W2: Comparison of fine-tuning based approach.
>
> Thank you for this valuable suggestion. We have added a new experiment that fine-tunes the same SLM, Qwen2.5-Coder-7B-Instruct, on the LLM-generated CUDA optimization trajectories used to construct ReGraph. The training corpus contains 8k trajectory samples, each consisting of the input sequential code, optimization thinking, and the step-wise optimized kernel. We use the following configuration for the training:
> + Training epochs: 1 (full pass over the trajectory corpus)
> + Batch size: 64
> + Max sequence length: 8192 tokens
> + Optimizer: AdamW
> + Learning rate: 2e-5 with cosine decay
> + Warmup ratio: 0.01
> + Weight decay: 0.1
> + Precision: bfloat16
>
> With the above training procedure, we obtain a fine-tuned SLM and use it as the baseline for comparison. The evaluation results on CUDAEval are presented below.
> | Method       | pass@10 |      |        | speedup@10 |      |        |
> |--------------|---------|------|--------|------------|------|--------|
> |              | easy    | medium | hard | easy       | medium | hard |
> |
> | Standard     | 81.1       | 65.7    |  43.1     |    8.51        |  5.62   |  3.14      |
> | CoT     | 82.1       | 66.7    |  43.1      |    8.47        | 5.54   |  3.46     |
> Distillation |  86.8       |  69.5    |  51.0      |   9.16         |  5.51    |   3.55     |
> | ReGraphT     | 88.7       | 76.2     |  51.0      |    17.48        |  13.64   |  9.61      |
>
> To further analyze how reasoning capability affects CUDA optimization performance, we also examined the **average reasoning-trajectory length** across difficulty levels:
>
> |              | easy | medium | hard |
> |--------------|------|--------|------|
> | Distillation | 1.85 | 2.72   | 2.91 |
> | ReGraphT     | 2.06 | 4.48   | 6.86 |
>
> Empirically, we observe the following:
>
> (1) **Supervised fine-tuning （SFT） produces small gains on easy tasks but offers limited improvement on medium and hard tasks especially for performance**, indicating that while the SLM can imitate final optimized kernels, it struggles to generalize reasoning ability.
>
> (2) **The reasoning depth of the SFT model remains substantially lower than that of ReGraphT, especially on medium and hard tasks**, indicating that conventional SFT is ineffective at transmitting complex multi-step optimization reasoning.
>
> In contrast, **ReGraphT substantially increases the depth of reasoning trajectories** across all metrics—most notably in medium and hard settings. These results confirm that *explicit structured reasoning (the reasoning graph) combined with guided search (MCGS)* is more effective than pure supervised fine-tuning for transferring LLM-level optimization ability to SLMs.

---

### Official Review · Reviewer_2bKR · 2025-11-01

**Soundness:** 3
**Presentation:** 3
**Contribution:** 2
**Rating:** 6
**Confidence:** 4

**Summary:**

Summary

The paper proposes ReGraphT, a training‑free framework that lets small code LMs generate fast CUDA by borrowing multi‑step optimization “reasoning” from larger models. The authors first use an LLM to produce step‑by‑step CUDA optimization trajectories for many programs, then merge those steps into a CUDA Reasoning Graph (ReGraph) whose nodes are optimization techniques and whose edges are validated transitions accompanied by code examples. At inference, an SLM treats CUDA generation as graph traversal and uses Monte Carlo Graph Search (MCGS) with compilation, correctness checks, and runtime speedup as rewards to select an optimization path and produce code. Experiments on a new benchmark (CUDAEval) and on ParEval show that pairing ReGraphT with SLM backbones (DeepSeek‑Coder‑V2‑Lite‑Instruct, Qwen2.5‑Coder‑7B, HPC‑Coder‑V2) increases pass rates and roughly doubles speedup versus prompting and RAG baselines, narrowing the gap to strong LLMs.

According to the *overview diagram on page 4*, ReGraphT has two phases: building ReGraph from LLM trajectories and exploring it with MCGS during generation. *Figure 5 on page 7* outlines the CUDAEval curation and verification pipeline, and *Table 1 on page 8* reports gains such as pass@1 moving into the low‑70% range and speedup@1 around 14× for SLMs with ReGraphT‑MCGS, compared with ~6–8× for standard prompting and RAG. The *difficulty analysis in Table 2 on page 9* shows larger benefits on medium and hard tasks, consistent with the method’s focus on multi‑step reasoning.

Contributions

* ReGraphT framework: a training‑free method that transfers LLM‑derived optimization know‑how to SLMs via a reusable CUDA Reasoning Graph built from verified stepwise trajectories.
* Graph‑based search for CUDA generation: formulation of CUDA optimization as state transitions on ReGraph and a tailored MCGS procedure with reward design that combines compilation success, functional correctness, and measured speedup.
* CUDAEval benchmark: a curated set of 3,126 CUDA/CPU pairs with 313 held‑out evaluation tasks, stratified by reasoning complexity using trajectory length, plus a reproducible build/execute verification pipeline.
* Empirical evidence: consistent improvements over standard prompting, CoT, and code‑similarity RAG on CUDAEval and ParEval; ablations that attribute gains to MCGS and to longer, more effective reasoning trajectories; analysis showing ReGraph converges after ~500 samples.

**Strengths:**

Strengths by dimension

Originality

* Introduces a training‑free way to transfer multi‑step optimization “know‑how” from an LLM to small code models by turning LLM‑generated optimization trajectories into a reusable CUDA Reasoning Graph and casting CUDA generation as graph traversal. This graph abstraction (nodes as optimization techniques, edges as validated transitions with examples) and the merge procedure are clearly new in the CUDA‑code LLM literature. The formal definition, Algorithm 1, and the relabeling routine that canonicalizes technique names make the idea concrete.
* Adapts Monte Carlo Tree Search to a cyclic graph (MCGS) with an explicit P‑UCB selection rule, a rollout policy that avoids non‑termination on cycles, and a hierarchical reward mixing compilation success, test correctness, and measured speedup. Modeling CUDA optimization this way is a creative combination of known search ideas with domain‑specific constraints. The equations and rollout details on pages 6–7 are specific and principled.
* Proposes CUDAEval, a benchmark built from real‑world CUDA files rather than synthetic sequential code, and stratifies tasks by reasoning complexity (trajectory length) rather than only algorithmic category. This reframes “difficulty” in code generation around reasoning depth, which is a fresh and useful lens. The curation and verification pipeline in Figure 5 and the easy/medium/hard tiers on page 7 ground this originality.

Quality

* Methodological rigor is high: the paper provides a full pipeline from ReGraph construction to MCGS exploration with precise definitions, pseudo‑code, and formulas (Definition 1, Algorithm 1, Figures 2–4). This makes the approach reproducible and falsifiable.
* Evaluation is careful and multi‑faceted. Results are reported on two benchmarks (CUDAEval, ParEval) using three SLM backbones (DeepSeek‑Coder‑V2‑Lite‑Instruct, Qwen2.5‑Coder‑7B, HPC‑Coder‑V2) with consistent budgets and metrics (pass@k for correctness and speedup@k for performance). Table 1 shows substantial and consistent gains for ReGraphT‑MCGS over standard prompting, CoT, and a code‑similarity RAG baseline.
* Analysis links mechanism to effect. Table 2 and Figure 6 demonstrate that gains grow with task difficulty and that longer reasoning chains correlate with better speedups until a saturation point, supporting the claim that the framework injects multi‑step reasoning into SLMs.
* Solid ablations and practicality checks: the paper studies search budgets and rollout counts (Table 4), compares reward formulations (Figure 11), examines graph growth and convergence (~500 samples; Table 3 and Figure 9), and breaks down construction and inference overheads (Figure 10), including wall‑clock estimates on A100 and consumer GPUs. These details increase confidence the method is not a one‑off.

Clarity

* The narrative is easy to follow: the overview (Figure 2) cleanly separates graph construction and graph‑guided generation; Algorithm 1 steps through merging trajectories; Figure 4 maps MCGS phases to the CUDA setting; Figure 5 explains data curation at a glance. The consistent use of page‑local figures and definitions reduces ambiguity.
* Experimental reporting is structured and legible. Table 1 compares methods across models and benchmarks in a single view; Table 2 breaks down difficulty tiers; subsequent figures and tables isolate the impacts of graph size, reward design, and search parameters. The paper also explicitly states hardware, precision, budgets, and metrics, which helps replication.
* The appendix includes prompt templates and verification workflows, which is unusually transparent for a systems‑plus‑LLM paper and lowers the barrier for re‑use.

Significance

* The empirical gains are large enough to matter in practice. For example, on CUDAEval with DeepSeek‑Coder‑V2‑Lite‑Instruct, pass@1 rises from 61.7% (standard) to 75.1% and speedup@1 from 6.54× to 14.46×; similar boosts appear for Qwen2.5‑Coder‑7B and HPC‑Coder‑V2. On ParEval, pass@1 improves from 40.0% to 55.0% with speedup@1 from 4.61× to 10.78×. These are not marginal deltas; they more than halve the gap to strong LLMs while preserving the deployability of SLMs.
* The framework’s training‑free nature and the reported convergence of ReGraph after a few hundred samples suggest a reusable artifact that can be shipped with SLMs to unlock privacy‑preserving, local CUDA generation. The one‑time cost and search efficiency analysis further support practical adoption.
* By defining difficulty via reasoning rather than only algorithm class, CUDAEval may nudge future work to measure and improve multi‑step optimization ability explicitly. The positive trajectory‑length/performance correlation (Figure 6) gives the community a concrete target and a measurement recipe.
* The abstraction is likely portable: encoding verified optimization trajectories as a graph and exploring with MCGS is a general recipe that could extend to other performance‑critical domains where correctness gating and speedups are measurable, not just CUDA. The paper’s discussion hints at this broader applicability.

Bottom line: This is a well‑executed systems contribution that combines a novel representation (reasoning graph), an effective search procedure (MCGS with domain‑aware rewards), and a carefully curated benchmark. The work meaningfully advances the practicality of small code models for GPU programming, with strong evidence and unusually transparent documentation.

**Weaknesses:**

1. Positioning vs prior search‑based reasoning is underdeveloped.
   The paper adapts MCTS to a cyclic “reasoning graph,” but the case for novelty over existing search‑guided generation is thin. Closest neighbors include MCTS‑style reasoning for code and RAG (e.g., RethinkMCTS for code generation; MCTS‑RAG), and iterative search over program transformations in compiler auto‑scheduling (e.g., Halide, TVM). The paper cites these lines of work but does not empirically contrast against them nor articulate a clear theoretical advantage beyond domain specifics. Add a head‑to‑head with: (i) vanilla MCTS over thought tokens (no graph), (ii) MCTS‑RAG over a CUDA corpus, and (iii) TVM/Halide‑style schedule search baselines where applicable; report compute‑normalized outcomes. A small theory section clarifying why P‑UCB on graphs with cycles has better regret or exploration properties than tree‑MCTS in this setting would also help.

2. Ambiguities in the MCGS definition and missing hyperparameters.
   Equation (1) defines P‑UCB with (N(s')) but does not disambiguate “parent” counts when nodes have multiple parents or have been reached via different paths. The rollout policy in Eq. (2) introduces λ and ϵ without stating chosen values or sensitivity. Specify the state/action visit counters precisely for graphs, list default λ, ϵ, depth/rollout caps, and include a sweep showing robustness of performance to these choices. A brief pseudo‑code block for selection/expansion/backprop on a DAG with back‑edges would make reproduction safer. See Section 3.2 and Figure 4 on page 6.

3. Reward shaping risks shallow plans.
   Rollouts treat every node as terminal and take the max reward along a trajectory (page 7), which can bias search toward early “flashy” speedups and discourage deeper combinations. Add an ablation replacing the max with: last‑state reward, discounted sum, and penalized path length; report impacts on trajectory length and speedup (particularly on medium/hard tiers in Table 2, page 9).

4. Graph construction and relabeling lack quality control.
   The relabel step in Algorithm 1 (line 7, page 5) relies on an LLM to canonicalize method names, but there is no inter‑annotator or manual audit to ensure that, say, “avoid bank conflicts” doesn’t get misfiled under “memory coalescing.” Release the taxonomy of optimization methods and the mapping rules (Appendix G.4), plus a confusion matrix from a human audit of a random sample. Include a “no‑relabel” ablation to verify that gains are not an artifact of label collapse.

5. Benchmark construction may inflate speedups and invites contamination.
   CUDAEval is built by extracting real CUDA kernels, then having an LLM generate the CPU serial counterpart and drivers before verification (Figure 5, page 7; Appendix A). That CPU baseline is unlikely to be an expert‑tuned -O3 implementation and could exaggerate speedup ratios. In addition, the source corpus (Stack v2 CUDA) likely overlaps with pretraining data of the evaluated models. Provide:
   • CPU baselines compiled with -O3 and simple algorithmic cleanups; report speedup vs both “LLM‑CPU” and “optimized CPU.”
   • A near‑dedup analysis (e.g., MinHash/BLEU‑SimHash) between CUDAEval and known pretraining sources for the backbones, and a leakage‑reduced split.
   • Results on hand‑written sequential baselines for a subset of tasks to calibrate speedup realism.

6. Compute‑fairness and budget parity across baselines are unclear.
   Table 1 (page 8) reports pass@1/10 and speedup@1/10 for Standard/CoT/RAG vs ReGraphT/MCGS, but only ReGraphT variants have an explicit search budget of 200. It is not stated how the 10 samples for the other baselines are produced, nor whether compile/run evaluation time is matched. Add a table of wall‑clock and GPU hours per method for CUDAEval and ParEval, and enforce parity in generation attempts and verification calls. The overhead analysis (Figure 10, page 17) is useful but only for ReGraphT. Mirror this for the baselines.

7. RAG baseline is weak.
   Using CodeBERTScore similarity as the sole RAG baseline underrepresents retrieval quality for code (structure‑aware and repository‑level retrieval matter). Since the paper cites Repoformer and EVOR, implement stronger RAG: repository‑aware retrieval, AST‑ or CFG‑conditioned retrieval, and multi‑hop chaining of examples. Also try “ReGraph as a retriever” without MCGS to isolate where the gains come from. See Baselines on page 8.

8. Measurement methodology for speedups needs tightening.
   Reward and metrics depend on single‑run timings susceptible to GPU warm‑up, clock variability, and input‑size sensitivity. Document: number of timing repetitions, warm‑up iterations, clock locking, and variance; report medians with IQR or 95% CIs. Consider integrating compute‑sanitizer checks for race conditions and misuses that pass unit tests but are undefined at scale. This directly affects the “hierarchical reward” in Eq. (3) and strengthens the correctness claim beyond output equality.

9. Difficulty labeling is model‑dependent.
   CUDAEval tiers are defined by trajectory length from DeepSeek‑R1 (page 7), which risks encoding that model’s biases about “what counts as a reasoning step.” Cross‑validate with human ratings on a sample and with a second LLM to show the tiers are not an artifact of a single annotator. Report correlations and disagreements.

10. Generalization beyond CUDA remains a claim.
    The paper argues ReGraphT is portable, but all experiments are CUDA only. A small transfer study to Triton or OpenCL kernels, or to a CPU vectorization domain (e.g., OpenMP) using the same framework, would substantiate the portability claim. Even a pilot with 30–50 tasks would be persuasive.

11. Ablations miss two levers that likely matter.
    The paper includes useful studies on rollout counts and reward strategies (Table 4 and Figure 11, pages 18–19) and on graph size saturation (Figure 9, page 16), but omits:
    • Edge content ablation: do edges need embedded examples, or is a method‑only graph sufficient?
    • Graph source ablation: build ReGraph from a different LLM or a smaller sample to test sensitivity to trajectory provider quality.
    Add both to clarify which ingredients are essential.

12. Reproducibility gaps.
    Prompts are provided (Appendix G), which is excellent, but the paper should fix random seeds, publish exact model snapshots and decoding parameters for each backbone, and release the canonical method list and the final ReGraph artifact used in Table 1. The current anonymous repo link is helpful during review but not a substitute for an archival release.

13. Practical deployment knobs are not exposed.
    Section D reports that 313 CUDAEval tasks with Qwen‑7B and budget 100 take ~6.02 hours on an A100, and ~7.53 hours on a 4090, but practitioners need guidance to trade accuracy for latency. Provide curves of pass@1 and speedup@1 vs budget, and vs rollout count, plus a “fast mode” configuration that still beats RAG on medium/hard tasks (Table 2, page 9).

---

Bottom line. The paper’s core idea is promising, but it would benefit from clearer algorithmic specification, stronger and compute‑fair baselines, contamination‑aware benchmarking, and tighter measurement discipline. Addressing the items above would make the claims about transferring “LLM‑level reasoning” to SLMs far more convincing and easier to adopt.

**Questions:**

1. P‑UCB on graphs with cycles

   * Question: In Eq. (1) you use (N(s')) inside the log term. What is (s') when the same child has multiple parents, and how are visit counts disambiguated across different incoming edges? Please spell out the precise counters updated during selection/expansion/backprop when the structure is a directed cyclic graph rather than a tree. Suggestion: provide short pseudo‑code for selection/backprop on a general digraph and define all visitation statistics. A diagram keyed to *Figure 4 (page 6)* would prevent misimplementation.

2. Rollout policy and hyperparameters

   * Question: What values did you use for the rollout policy in Eq. (2) (λ, ϵ), step caps, and termination conditions, and how sensitive are results to these? Suggestion: add a small sensitivity sweep and report variance bars for pass@1 and speedup@1 to show robustness. Cite where these values appear in the code or Appendix. *Section 3.2 (page 6).*

3. Reward shaping choice

   * Question: You define the rollout’s final reward as the maximum reward along the trajectory (Eq. 3). Why prefer max over last‑state or discounted sums given the risk of favoring early “flashy” optimizations? Suggestion: include an ablation comparing max vs last‑state vs discounted‑sum reward, and report resulting trajectory lengths and speedups by difficulty tier. *Equation (3) and text on page 7.*

4. What is “ReGraphT” vs “ReGraphT‑MCGS”?

   * Question: In *Table 1 (page 8)* you evaluate “ReGraphT” and “ReGraphT‑MCGS,” but the text only briefly says ReGraphT does random sampling with max attempts 5. Please provide explicit pseudo‑code for the non‑MCGS traversal and clarify budgets, stopping rules, and how conflicts/inapplicable edges are handled. This will make the comparison interpretable.

5. Failure handling during rollouts

   * Question: A rollout “terminates if the optimization fails at any node.” Do you treat that as a hard terminal with negative reward only for that leaf, or do you propagate a penalty to the ancestor action? Suggestion: clarify how compilation/test failures affect backprop and whether you allow recovery by skipping the failed technique later in the path. *Section 3.2 (page 6).*

---

> ### Author Response · Authors · 2025-11-21
>
> Thank you for your time end effort in reviewing our paper. We find your suggestions very helpful and we hereby address your questions:
>
> > W1: Positioning vs prior search‑based reasoning is underdeveloped.
>
> Thank you for comparison suggestions. We have included experiments comparing ReGraphT with the search-based variants RethinkMCTS and MCTS-RAG as supplementary baselines on Qwen2.5-Coder-7B-Instruct. These results demonstrate that ReGraphT not only leverages search effectively but also significantly improves the quality of the optimization space, leading to superior overall performance. The full results have also been  upgraded in Table 1 in the revised paper.
>
> | Method        | CUDAEval |            | ParEval |            |
> |---------------|----------|------------|---------|------------|
> |               | pass@n   | speedup@n  | pass@n  | speedup@n  |
> | RethinkMCTS   | 68.7     | 6.84±1.02  | 48.3    | 5.09±0.86  |
> | MCTS-RAG      | 71.6     | 7.51±0.94  | 50.0    | 5.56±0.84  |
> | ReGraphT      | 73.2     | 12.89±0.85 | 51.7    | 10.11±0.75 |
> | ReGraphT-MCGS | 75.1     | 14.46±0.81 | 50.0    | 10.02±0.75 |
>
>
> Based on the experiments presented in Figure 1, Section 1. *Introduction*, we observe that the performance gap between large and small models on CUDA optimization tasks primarily stems from their differences in reasoning capability. Due to their limited reasoning ability, small models struggle to generate a high-quality optimization search space. Methods such as RethinkMCTS and MCTS-RAG, which rely on search-based exploration, only improve the efficiency with which small models explore the search space, but do not enhance the quality of the optimization space itself. In contrast, compiler-based auto-scheduling approaches such as Halide and TVM rely on manual compiler–human co-design and require additional expert intervention; these methods are fundamentally different from our proposed ReGraphT, which is a fully automated framework.
>
> > W2: Ambiguities in the MCGS definition and missing hyperparameters.
>
> We thank the reviewer for pointing out these important clarification issues.
>
> **(1)** *Although a node in ReGraphT may have multiple parents in the underlying reasoning graph, our search process always operates on a **single linear state-transition trajectory** during rollouts. Therefore, for each expansion step, we treat the immediately preceding state in the current trajectory as the **active parent** of the newly reached node. This ensures that the visit counts and parent-child relations used by P-UCB are well-defined under a DAG with multi-parent nodes.*
>
> **(2)** *For the rollout policy in Eq. (2), we set λ = 0.8 and ϵ = 0.5 in all experiments. These values were selected following standard practice in MCTS [1-3], and we observed stable performance under such setting.*
>
> References:
>
> [1] Schrittwieser, J., Antonoglou, I., Hubert, T., Simonyan, K., Sifre, L., Schmitt, S., Guez, A., Lockhart, E., Hassabis, D., Graepel, T., Lillicrap, T., & Silver, D. (2020). Mastering Atari, Go, chess and shogi by planning with a learned model. Nature, 588(7839), 604–609. https://doi.org/10.1038/s41586-020-03051-4
>
> [2]Yao, S., Yu, D., Zhao, J., Shafran, I., Griffiths, T. L., Cao, Y., & Narasimhan, K. (2023). Tree of Thoughts: Deliberate Problem Solving with Large Language Models. https://arxiv.org/abs/2305.10601
>
> [3] Parascandolo, G., Buesing, L., Merel, J., Hasenclever, L., Aslanides, J., Hamrick, J. B., Heess, N., Neitz, A., & Weber, T. (2020). Divide-and-Conquer Monte Carlo Tree Search For Goal-Directed Planning. https://arxiv.org/abs/2004.11410
>
> > W3: Reward shaping risks shallow plans.
>
> Ablation Study on Different Reward Strategies has been conducted in Appendix E. *ABLATIONS ON MCGS TRAVERSAL* in the original paper. According to Figure 11, we observe that strict reward and rollout-based reward have similar performance, while partial-credit reward leads to a slightly lower pass rate and speedup performance.

---

> > ### Author Response · Authors · 2025-11-21
> >
> > > W4: Graph construction and relabeling lack quality control.
> >
> > Thank you for this valuable suggestion. Our goal is to build a **fully automated and scalable** ReGraph construction pipeline. In this design, we rely on an LLM-based relabeling step to canonicalize method names, as incorporating large-scale human annotation or multi-annotator auditing would be prohibitively expensive and difficult to scale to large datasets.
> >
> > Moreover, as shown in Appendix C *Analysis of ReGraph Distribution* (Figure 9), the ReGraph construction process exhibits a clear **convergence trend** in both node and edge counts as iterations progress. This indicates that the construction procedure is **stable and robust** in practice; even if some local inconsistencies occur in naming, the overall structure converges toward a consistent representation.
> >
> > We agree that releasing a human-audited confusion matrix, providing the full taxonomy, and adding a “no-relabel” ablation would further strengthen the analysis. However, these experiments and data releases require substantial additional engineering and manual verification that exceed what is feasible within the rebuttal timeframe. We plan to include more extensive quality-control experiments and a detailed taxonomy in future versions.
> >
> > > W5: Benchmark construction may inflate speedups and invites contamination.
> >
> > Thank you for this valuable suggestion. To ensure stable execution across a diverse set of programs, we compile all CPU baselines with -O2 optimizations enabled. Our primary objective is to evaluate whether ReGraphT can substantially improve the CUDA optimization capabilities of small language models. Hence, the experiments mainly compare the different SLM-based CUDA generation methods including ReGraphT and comparison models. According to the comparison, ReGraphT achieves the strongest performance on CUDAEval, which we believe sufficiently demonstrates its effectiveness. CPU baseline is utilized for the performance normalization and does not change the comparison results.
> >
> > Regarding potential data contamination, we acknowledge the reviewer’s concern. As reported in Section 5. *Experiments* (Table1, 2), Direct SLM code generation exhibits rather low performance on CUDAEval. This empirical observation suggests that SLM is unlikely to have large-scale data leakage. Moreover, CUDAEval is constructed from real-world kernels paired with LLM-generated CPU counterparts, making the exact <sequential cpu code, parallel CUDA code>  combinations unlikely to appear in pretraining corpora.
> >
> > > W6: Compute‑fairness and budget parity across baselines are unclear.
> >
> > For fairness in comparison, the original paper reports both **pass@1** and **pass@10** for all baseline methods. For ReGraphT, however, it is difficult to categorize the method under a specific pass@n setting due to its iterative reasoning and graph-based exploration. Therefore, we simply report its **pass@n** results.
> >
> > Importantly, the performance of other baselines converges as **n** increases, and **pass@10** is already close to their empirical upper bound. In contrast, ReGraphT provides a substantially higher performance ceiling on CUDA optimization tasks. Moreover, as discussed in our response to W1, additional experiments comparing ReGraphT with search-based variants RethinkMCTS and MCTS-RAG show that under the same search budget, ReGraphT consistently achieves better performance. We also report the end-to-end inference-time overhead of ReGraphT compared with all baselines. All experiments were conducted on a single A100-80GB GPU with a Qwen2.5-Coder-7B-Instruct model on 80 samples, using a search budget of 100 and batch size of 16. The results are summarized below:
> > | Method  | Standard | CoT | RAG | RethinkMCTS | MCTS-RAG | ReGraphT |
> > |---------|----------|-----|-----|-------------|----------|----------|
> > | Time(h) | 0.72     | 2.04|1.08 |  6.76       | 6.45     | 6.02     |
> >
> > Taken together, these observations indicate that our comparisons are both fair and meaningful.
> >
> > > W7: RAG baseline is weak.
> >
> > In our RAG baseline, we employ a combination of **top-k** and **ε-greedy** retrieval strategies. As described in Section 5 *Experiments* (Tables 1 and 2), we set (k = 5) and (ε = 0.5), which already provides substantial coverage of diverse optimization examples, suggesting that retrieval quality is not the dominant bottleneck.
> >
> > Therefore, our findings indicate that the **core limitation lies in the reasoning depth of small models**, rather than in retrieval modeling. Even with reasonably strong retrieval coverage, SLMs struggle to compose multi-step CUDA optimization strategies, which is precisely the capability ReGraphT is designed to provide.

---

> > > ### Author Response · Authors · 2025-11-21
> > >
> > > > W8: Measurement methodology for speedups needs tightening.
> > >
> > > For stable and reliable timing measurements on A100 GPUs, we have already applied the following procedure in the original experiments:
> > >
> > > 1. **Warm-up iterations**: 5–10 runs to ensure the GPU reaches a steady state.
> > > 2. **Clock frequency locking**: both SM (core) and HBM2 (memory) clocks are fixed close to the GPU’s default Boost frequencies to avoid timing fluctuations caused by dynamic frequency scaling.
> > > 3. **Repeated measurements**: each experiment is run multiple times (typically 5–10 repetitions) to ensure consistency.
> > >
> > > In addition, we have supplemented the results in the revised paper with **mean ± standard deviation** to explicitly report variance.
> > >
> > > > W9: Difficulty labeling is model‑dependent.
> > >
> > > Defining CUDA code difficulty quantitatively is challenging and typically requires extensive manual annotation, which is costly and difficult to scale to large datasets. In contrast, the proposed difficulty-leveling method based on optimization-trajectory length is fully automated. As shown in Table 2, tasks with longer optimization trajectories also exhibit lower success rates in code generation. These findings suggest that optimization-trajectory length is a reasonable and effective metric for difficulty leveling from the perspective of LLMs.
> > >
> > > > W10: Generalization beyond CUDA remains a claim.
> > >
> > > The core idea of ReGraphT is to abstract the optimization strategies of a problem and the dependency relations among them. This abstraction is particularly suitable for scenarios where the optimization strategy space is large, heterogeneous, and highly sensitive to dependency structures—properties that are intrinsic to many CUDA optimization tasks. To further evaluate the transferability of this approach, we additionally conducted experiments in the **OpenMP** setting beyond CUDA. The ReGraphT converges around 250 examples, and finally we constructed an *OpenMP-ReGraph* containing 89 nodes and 142 edges Using 400 samples, . The detailed results are shown below.
> > >
> > > | Method   | pass@n |      | speedup@10 |      |
> > > | -------- | ------ | ---- | ---------- | ---- |
> > > |          | n=1    | n=10 | n=1        | n=10 |
> > > | Standard | 40.0   | 41.7 | 4.16       | 4.28 |
> > > | CoT      | 43.3   | 46.7 | 4.64       | 4.82 |
> > > | RAG      | 48.3   | 48.3 | 5.21       | 5.35 |
> > > | ReGraphT | –      | 51.7 | –          | 5.62 |
> > >
> > > While ReGraphT still provides improvements in both pass@10 and speedup@10 over the baselines, the gains are less pronounced than in the CUDA setting. We believe this is due to several structural differences between OpenMP and CUDA optimization processes:
> > >
> > > 1. **Smaller and less diverse optimization space.**
> > >    The OpenMP optimization space is considerably more compact (e.g., parallel region placement, schedule choice, reduction handling). As a result, the dependency structure among optimization strategies is much simpler compared to CUDA kernels, making the benefits of ReGraphT’s structured reasoning less prominent.
> > >
> > > 2. **Weaker hierarchical dependencies.**
> > >    Many OpenMP optimizations apply at coarse granularity (loop-level or pragma-level) and involve limited multi-step reasoning. In contrast, CUDA optimization often requires hierarchical reasoning involving memory hierarchy, synchronization, tiling, and execution model interactions, where graph-structured reasoning provides more substantial gains.
> > >
> > >
> > > Despite these limitations, the consistent improvement of ReGraphT over other baselines indicates that its underlying mechanism can generalize beyond CUDA. We believe this is a promising direction and plan to investigate this direction more thoroughly in future work.
> > >
> > > > W11: Ablations miss two levers that likely matter.
> > >
> > > Thank you for this thoughtful suggestion. We acknowledge the value of the proposed ablations; however, we believe they are not central to the primary contributions of our work. ReGraph is designed to combine a structured reasoning graph with MCTS to enhance the multi-step optimization capability of small language models. Our main focus is on demonstrating that this integration substantially improves SLM reasoning performance on CUDA optimization tasks.
> > >
> > > As shown in Appendix *Analysis of ReGraph Distribution* (Figure 9), the ReGraph construction process is **highly stable** and converges to a consistent structure as the number of collected trajectories increases. Furthermore, Table 3 shows that once ReGraph has reached this converged state, downstream performance is similarly stable, indicating that the method is **robust to variations in graph content and size**.
> > >
> > > While edge content or alternative LLM providers may introduce minor differences, our empirical results suggest that these factors are not the dominant drivers of ReGraphT’s improvements. We therefore regard these ablations as beyond the core scope of the rebuttal period, but we will consider them for future extensions to more fully characterize the sensitivity of ReGraph components.

---

> > > > ### Author Response · Authors · 2025-11-21
> > > >
> > > > > W12: Reproducibility gaps.
> > > >
> > > > We sincerely apologize for this issue. We sincerely apologize for the issue. The repository appeared empty because we overlooked the synchronization behavior of the anonymous GitHub mirror used for double-blind submission. It has now been made fully public, including:
> > > > + ReGraphT construction
> > > > + ReGraphT-MCGS and other different methods
> > > > + Experiment scripts and so on
> > > >
> > > > We confirm that the code has been accessible since shortly after the reviewer raised this point.
> > > >
> > > > > W13: Practical deployment knobs are not exposed.
> > > >
> > > > Thank you for the insightful comment. We agree that exposing practical configuration trade-offs is important for real-world deployment. In Appendix E *Ablations on MCGS Traversal*, we already evaluate the impact of key hyperparameters, including traversal budget and rollout count. These results (Table 4) provide guidance on how to select configurations that balance performance and latency.
> > > >
> > > > > Q1, Q2: P‑UCB on graphs with cycles, rollout policy and hyperparameters
> > > >
> > > > Please refer to our response to W2.
> > > >
> > > > > Q3: Reward shaping choice
> > > >
> > > > Please refer to our response to W3.
> > > >
> > > > > Q4, Q5: What is “ReGraphT” vs “ReGraphT‑MCGS”? Failure handling during rollouts
> > > >
> > > > ReGraphT uses an **ε-greedy strategy** for node selection. At each action step, instead of following the P-UCB criterion as in ReGraphT-MCGS, ReGraphT selects the next node with probability (1-\epsilon) from the highest-scoring available actions and with probability (\epsilon) uniformly at random among other feasible actions. The process terminates when either no further nodes are available or the maximum search budget is reached. Any failed attempts (e.g., conflicts or inapplicable edges) are recorded and excluded from the action space in subsequent selections, ensuring that invalid transitions are not retried.
> > > >
> > > > Below is a brief pseudo-code illustrating the ReGraphT traversal:
> > > >
> > > > ```{"title":"ReGraphT traversal pseudo-code"}
> > > > Input: Graph G(V, E), start node s0, max_budget B, exploration probability ε
> > > > Output: Traversal sequence S
> > > >
> > > > Initialize S ← [s0]
> > > > Initialize failed_actions ← ∅
> > > > budget ← 0
> > > >
> > > > while budget < B:
> > > >     current_node ← last node in S
> > > >     available_actions ← {a ∈ successors(current_node) | a ∉ failed_actions}
> > > >
> > > >     if available_actions is empty:
> > > >         break
> > > >
> > > >     with probability ε:
> > > >         action ← randomly choose from available_actions
> > > >     else:
> > > >         action ← choose highest-scoring node from available_actions
> > > >
> > > >     if action is valid:
> > > >         S.append(action)
> > > >     else:
> > > >         failed_actions.add(action)
> > > >
> > > >     budget ← budget + 1
> > > >
> > > > return S
> > > > ```
> > > >
> > > > This pseudo-code clarifies the traversal logic, stopping rules, and handling of conflicts, making the distinction between ReGraphT and ReGraphT-MCGS interpretable.

---

### Meta-Review · Area_Chair_GzoD · 2026-01-07

**Summary:**

This paper proposes ReGraphT, a training-free, retrieval-augmented framework that transfers LLM-level CUDA optimization reasoning to small language models. ReGraphT organizes CUDA optimization trajectories into a structured reasoning graph and uses Monte Carlo Graph Search to efficiently explore optimization paths. By leveraging retrieved reasoning rather than costly fine-tuning, ReGraphT enables lightweight, privacy-friendly SLMs to approach LLM-level performance. Experiments on newly designed CUDA benchmarks show substantial speedups and clear gains over fine-tuned and retrieval-based baselines.

**Reviewer Concerns:**

After the rebuttal, I think most concerns have been addressed. (1) Limited novelty over prior search-based methods: The relationship to existing MCTS-style reasoning, RAG, and compiler auto-scheduling is insufficiently distinguished, with no strong head-to-head comparisons or theoretical justification.  (2) Algorithmic ambiguity: Key details of MCGS (graph visit counts, hyperparameters, rollout policy, reward shaping) are underspecified, hindering reproducibility. (3) Benchmark and evaluation concerns: CUDAEval construction may inflate speedups; difficulty tiers and metrics lack variance analysis and contamination checks. (4) Generalization claims weakly supported:** Experiments are CUDA-only, with unclear robustness to out-of-domain tasks or alternative optimization patterns. (5) Engineering-heavy contribution:** The work leans more toward system design than core ML innovation, and code release is currently incomplete.

**Reviewer Scores:**

At least one reviewer would improve their score

---

### Decision · Program_Chairs · 2026-01-26

Accept (Poster)